# Intracellular artificial supramolecules based on de novo designed Y15 peptides

Takayuki Miki [1✉], Taichi Nakai[1], Masahiro Hashimoto [1], Keigo Kajiwara[1], Hiroshi Tsutsumi[1] & Hisakazu Mihara [1]

De novo designed self-assembling peptides (SAPs) are promising building blocks of supramolecular biomaterials, which can fulfill a wide range of applications, such as scaffolds for tissue culture, three-dimensional cell culture, and vaccine adjuvants. Nevertheless, the use of SAPs in intracellular spaces has mostly been unexplored. Here, we report a self-assembling peptide, Y15 (YEYKYEYKYEYKYEY), which readily forms β-sheet structures to facilitate bottom-up synthesis of functional protein assemblies in living cells. Superfolder green fluorescent protein (sfGFP) fused to Y15 assembles into fibrils and is observed as fluorescent puncta in mammalian cells. Y15 self-assembly is validated by fluorescence anisotropy and pull-down assays. By using the Y15 platform, we demonstrate intracellular reconstitution of Nck assembly, a Src-homology 2 and 3 domain-containing adaptor protein. The artificial clusters of Nck induce N-WASP (neural Wiskott-Aldrich syndrome protein)-mediated actin polymerization, and the functional importance of Nck domain valency and density is evaluated.

[1] School of Life Science and Technology, Tokyo Institute of Technology, Yokohama, Kanagawa, Japan. ✉email: tmiki@bio.titech.ac.jp

In natural biological systems, non-covalent interactions drive macromolecules to spontaneously organize into supramolecular structures, giving rise to complex functionality. For example, cytoskeleton components, such as actin filaments and microtubules, which play central roles in cell shaping, migration, and intracellular transportation, form in this manner[1]. Therefore, platform techniques that build artificial protein assemblies in living systems are crucial for achieving the ultimate goal of synthetic biology: elucidating biological systems through mimicry and generating systems with artificial functions[2,3].

Various de novo designed peptides have been developed to produce valuable and practical supramolecular biomaterials in bottom-up strategies. In this approach, peptide-based supramolecular structures are created from scratch, which takes into account extensive interactions such as van der Waals forces, hydrogen bonds, and electrostatic and aromatic interactions. As a representative example, supramolecular nanofibers composed of designed self-assembling peptides (SAPs) are attractive biomaterials for tissue engineering, wound healing, and vaccines for the following two reasons[4,5]: First, biomaterials can provide synthetic scaffolds for these applications. Amphiphilic peptides, composed of an alternating sequence of hydrophobic and hydrophilic amino acids, favor the formation of β-sheet assemblies in an aqueous solution. As representative examples, FKE12[6], MAX-I[7–9], and RADA-16[10] fabricate fibrils, and these networks lead to hydrogelation. Similarly, our group has also exploited the Y9 self-assembling peptide[11–14]; second, desired functions can be rationally created by linking bioactive molecules to SAPs. For instance, Collier and colleagues developed a 'βTail' peptide, which enables the installation of functional proteins into nanofibers, and the dose of the displayed molecules can be precisely tuned[15].

As shown in biomaterial development, de novo peptides provide promising building blocks for constructing artificial supramolecular assembly in living cells. Despite these advantages, few attempts to apply these SAP-based materials to biological tools in intracellular spaces have been reported. While pioneering reports from Xu and colleagues have described intracellular enzyme sequestration by introducing precursors of hydrogelators consisting of a synthetic peptide containing a ligand of an enzyme[16,17], intracellular assembly of genetically encoded SAPs composed of natural amino acids has been largely unexplored. Unlike the self-assembly of synthesized peptides in test tubes, the design and development of SAPs interacting in cells are challenging owing to the fact that the cell environment is a crude and crowded milieu where macromolecules occupy ~30% of the total volume[18]. Furthermore, genetically expressed substrates are present in mammalian cells at low concentrations (on the order of nM–μM), while the assembly properties of synthesized peptides can be assessed at higher concentrations (on the order of 0.1 wt%).

Here, we report a de novo peptide, Y15 (YEYKYEYKYEY-KYEY), which displays a high propensity to assemble in cellular contexts. In this design, Y15 forms an amphiphilic β-strand, with a hydrophobic Tyr residue on one side and hydrophilic Glu and Lys residues on the other. Thus, adjacent β-strands can interact through hydrophobic and π–π interactions between Tyr residues as well as electrostatic interactions between negatively charged Glu and positively charged Lys residues. Y15 forms β-sheet nanofibers in an aqueous solution, and by genetic tagging of model protein superfolder green fluorescence protein (sfGFP) with Y15, we found that it could assemble into fibrous structures in test tubes. The clustering of Y15-sfGFP in living cells was validated by fluorescence microscopy, homo-Förster resonance energy transfer (FRET) analysis, and pull-down assays. We could also robustly generate artificial microscale structures by fusing Y15 to Azami-Green (AG), a tetrameric fluorescent protein.

Furthermore, Y15-based assemblies can be decorated with functional proteins. To demonstrate its intracellular application, we reconstituted Nck clusters by introducing Y15-tagged Nck(1–258) into Y15-based supramolecules, leading to N-WASP-mediated actin polymerization. From in-cell reconstitution studies, the contribution of domain valency and the density-dependency on function were evaluated in situ. Taken together, this study demonstrates that our de novo designed Y15 peptide offers a tool for organizing supramolecular architectures with biological functions in living cells.

## Results

**Self-assembly of Yn peptides**. Natural proteins interact with large contact surfaces for selective molecular recognition in complex cellular environments. As such, the development of de novo short peptides with high potential to self-assemble at low concentrations (on the order of μM) is incredibly challenging. To enhance the self-assembling propensity, we used a simple strategy involving elongation of the peptide length. Using the YE and YK units of the Y9 peptide[11–14], we designed and synthesized Yn peptides with four different lengths—Y9, Y11, Y13, and Y15—and the negative control peptide Y15(K9P), in which Lys in the middle of Y15 is substituted with Pro, disfavoring β-sheet formation (Fig. 1a, b, and Supplementary Fig. 1). Peptide assembly was monitored by enhancement in thioflavin T (ThT) fluorescence upon binding to amyloid-like fibrils (Fig. 1c). These peptides were dissolved at concentrations of 5–50 μM in phosphate-buffered saline (PBS), and the solution was incubated for 24 h at 37 °C. Y13 and Y15 showed strong fluorescence over the whole concentration range, whereas Y11 only enhanced the fluorescence above 20 μM, and Y9 did not show significant fluorescence even at 50 μM (Supplementary Fig. 2). In the circular dichroism (CD) spectrum, Y15 showed a large negative band at ~226 nm ($[\theta]_{226} = -6.2 \pm 1.0 \times 10^4 \deg \, cm^2 \, dmol^{-1}$) at 10 μM (Supplementary Fig. 3). Compared with typical spectra of β-sheets, the peak wavelength was red-shifted, and the intensity was relatively high. We attributed these bands to the ordered array of the aromatic side chains of Tyr residues and/or twisted β-sheets[19,20]. The negative Cotton effect became weaker as the peptide shortened. Fourier transform infrared (FT-IR) spectra were recorded to determine secondary structure, with the Y15 peptide showing a peak between 1610 and 1630 cm$^{-1}$, indicating a typical β-sheet structure (Supplementary Fig. 3). As shown by negative stain transmission electron microscopy (TEM), Y15 formed nanofibers with a width of 5.0 ± 1.2 nm, consistent with the length of a stretched peptide (Fig. 1d and Supplementary Fig. 3). Y11 and Y13 peptides also adopted fibril structures but no structure was observed for the shortest Y9 peptide (Fig. 1d). These results show that Y15 self-assembles readily at low concentrations (5 μM) and forms nanofiber structures with a β-sheet secondary structure.

Some intrinsically disordered regions undergo liquid-liquid phase separation (LLPS) and subsequently form fibril structures[21–23]. To investigate whether Y15 self-assembles through phase separation, we examined the time profile of thioflavin-T fluorescence intensity and observed the phase behavior by differential interference contrast (DIC) microscopy (Supplementary Fig. 4). Surprisingly, enhancement of thioflavin-T fluorescence occurred without a lag time and plateaued within 3 min. In contrast, no structure was observed initially; then thioflavin-T-positive fibrillar aggregates arose and elongated. The time required for aggregate formation was concentration dependent. The aggregate grew and precipitated when heated to 80 °C. Because liquid droplets were not detected at any incubation time, we expected Y15 to self-assemble rapidly into fibrils that gradually entangle to form aggregates. TEM observations revealed that the microscale aggregates consisted of entangled fibers (Supplementary Fig. 3).

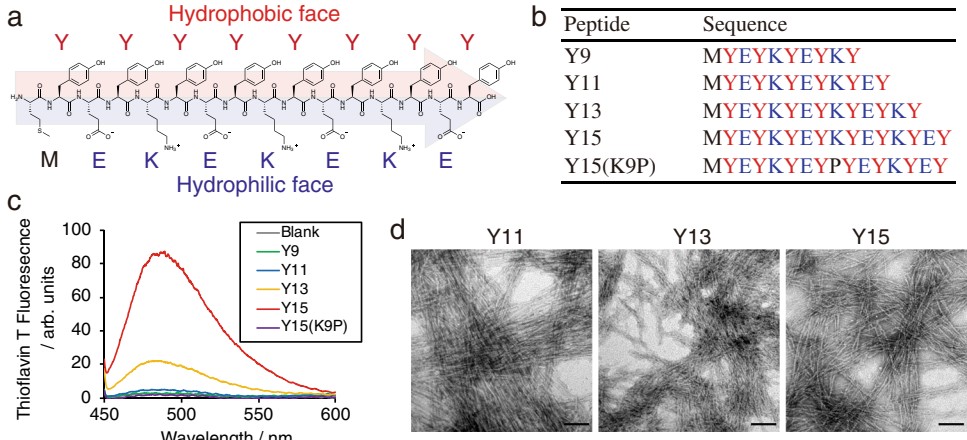

**Fig. 1 Self-assembly of Yn de novo peptides. a** Chemical structure of Y15 peptide. **b** A list of tested Yn peptides that differ in length and the negative control. **c** Thioflavin T fluorescence assay of Yn peptide assembly. Yn peptides (10 μM) in PBS were incubated for 24 h at 37 °C and then 25 μM thioflavin T dye was added. The fluorescence spectra were measured by F-7000 (Hitachi) with FL Solution. Source data are provided as a Source data file. **d** Negative-stained TEM observations of Y11, Y13, and Y15. Peptides (100 μM) in PBS were incubated for 22 h at 37 °C. The mixture was attached to grids and negatively stained by nano-W. Scale bar, 100 nm.

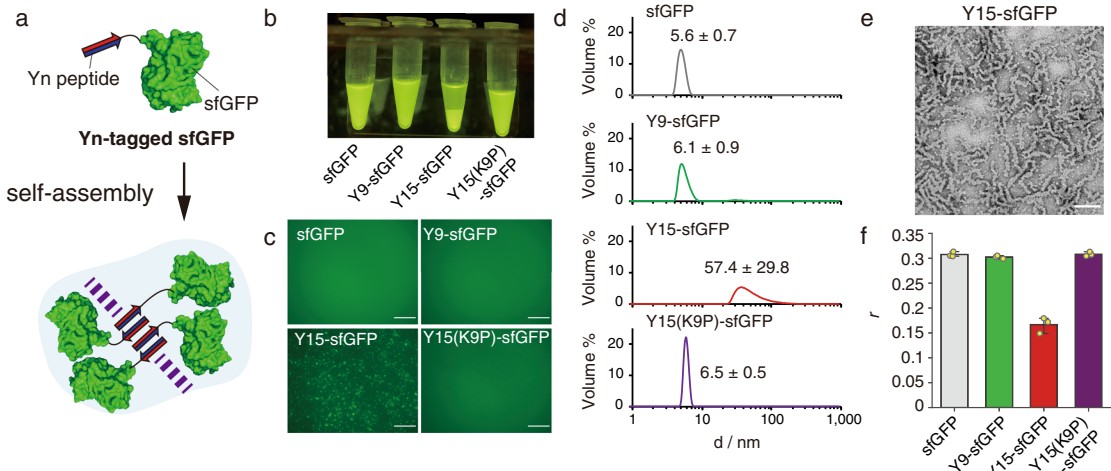

**Fig. 2 Self-assembly of Y15-fused sfGFP in test tubes. a** Schematic illustration of Yn-tagged sfGFP self-assembly. **b** A fluorescence optical image of Yn-sfGFP solutions. The 15 μM dialyzed proteins in 10 mM phosphate buffer (pH 7.2) with 1 mM EDTA were incubated at 4 °C, resulting in formation of fluorescent Y15-sfGFP condensates. **c** Fluorescence microscopy images of protein solutions (1 μM) in PBS. Scale bar, 100 μm. **d** DLS measurements of 5 μM protein solutions in 10 mM phosphate buffer (pH 7.2). The protein solutions were measured by ZS90 (Malvern) with Zetasizer Software. Y15-sfGFP formed clusters in solution, whereas the other proteins were monomers. **e** Negative-stained TEM image of Y15-sfGFP. Scale bar, 100 nm. Curly fibrils with a uniform width (13.7 ± 3.2 nm) were observed. **f** Fluorescence anisotropy assay of 5 μM protein solutions. Homo-FRET of sfGFP in Y15-sfGFP clusters showed a decrease in fluorescence anisotropy ($r$). Data are presented as mean values $+/-$ SD ($n = 3$ biologically independent experiments). Source data are provided as a Source Data file.

Although the net charges of the Yn peptides at physiological pH differed (Y9 and Y13, neutral; Y11 and Y15, −1), their ability to self-assemble was easily improved by increasing the length. Y15 assembly was less susceptible to ionic strength below 500 mM NaCl (Supplementary Fig. 5). We hypothesize that electrostatic interactions between Glu and Lys side chains contribute less to assembly than hydrogen bonds on the main chain and the hydrophobic/aromatic interactions of Tyr on the hydrophobic face.

**Self-assembly of Yn-tagged sfGFPs.** Given the strong self-assembling ability of the Y15 peptide, we hypothesized that Y15 could be a core motif to form protein assemblies in liquid solution (Fig. 2a). To gain insights into the assembly behavior of Yn-tagged proteins, we chose sfGFP[24] as a model protein and designed four Yn-tagged sfGFPs (Y9-, Y11-, Y13-, and Y15-sfGFP) in which Yn peptides of different lengths were genetically fused to the N-terminus of sfGFP via a Gly linker (G3). In addition, we used two negative controls: sfGFP without a tag and Y15(K9P)-sfGFP. These proteins were expressed in *Escherichia coli* and purified by TALON® resin (Supplementary Fig. 6). The Yn-tagged sfGFPs were extracted from inclusion bodies and refolded after purification. The 15 μM Y13- and Y15-sfGFP solutions gradually separated into two phases over 2 weeks of incubation at 4 °C, while the Y9- and Y11-sfGFP solutions and other control solutions were single-phase and transparent (Fig. 2b and Supplementary Fig. 6). At 1 μM, Y13- and Y15-sfGFP formed fluorescent microscale structures in PBS (Fig. 2c). The fluorescent puncta increased in the presence of 10% dextran, a molecular crowding agent (Supplementary Fig. 6). However, the

short peptide fusions Y9- and Y11-sfGFP and the negative controls were completely dissolved.

All variants were obtained in soluble form in 10 mM sodium phosphate buffer (pH 7.2) at a 5-μM concentration. Dynamic light scattering (DLS) measurements showed cluster formation of Y13-sfGFP ($58 \pm 22$ nm) and Y15-sfGFP ($57 \pm 30$ nm), while the shortest peptide, Y9-sfGFP, and the negative controls had a diameter consistent with that of the sfGFP monomer (Fig. 2d and Supplementary Fig. 7). To explore the morphology of the Y15-sfGFP assembly, we carried out negative stain TEM observation (Fig. 2e), revealing fibrils $13.7 \pm 3.2$ nm in width, which is consistent with the distance between the edges of the opposite sfGFPs in the antiparallel model (Supplementary Fig. 8). In contrast, synthetic Y15 peptides formed straight, long fibrils whose ends could not be determined by TEM analysis. The Y15-sfGFP fibrils were curly and the length ($119 \pm 71$ nm) was finite, presumably because of frustrated growth[25] driven by the steric constraints of sfGFPs.

Homo-FRET measurements[26], which were observed as a decrease in fluorescence anisotropy ($r$), showed a twofold lower $r$-value of Y15-sfGFP than sfGFP, supporting cluster formation (Figs. 2f and S9). The critical aggregation concentrations (CAC) of Y13-sfGFP and Y15-sfGFP were determined by the change in fluorescence anisotropy to be ~300 and 50 nM, respectively (Supplementary Fig. 9). We conclude that the strong assembly propensity of the Y13 and Y15 peptides enables protein assembly.

**Self-assembly of the Y15-tagged protein in living cells.** As a proof-of-concept of self-assembly in living cells, Yn-tagged sfGFPs were expressed in HEK293 cells and observed by confocal fluorescence microscopy (Fig. 3a). We noted clear fluorescent puncta in the Y15-sfGFP-expressing cells. Y13-sfGFP was mainly distributed throughout cells, and fluorescent granules were also observed in highly expressed cells. In contrast, sfGFP without a tag and Y9-, Y11-, and Y15(K9P)-sfGFP were distributed evenly throughout the whole cell. Figure 3b shows the partition coefficients, indicating that Y15 is required for protein assembly in cells. To quantify the proportion of assembled sfGFPs, we lysed cells and fractionated them by sedimentation. Y15-sfGFP was more abundant in the pellet ($62 \pm 7\%$) than Y13-sfGFP ($45 \pm 4\%$), and the vast majority of other controls remained in a soluble form (Fig. 3c). Y15-sfGFP showed half the fluorescence anisotropy of sfGFP, which is consistent with the test tube experiments, suggesting cluster formation in living cells (Fig. 3d). On the basis of the finding that assembly was dependent on peptide length, we determined that the Y15 peptide retains sufficient self-assembling propensity in living cells.

To validate the Y15-Y15 interaction in living cells, we co-expressed Y15-sfGFP and Y15-mCherry-HA tag in HEK293 cells. Both proteins were highly co-localized (Pearson $R = 0.96 \pm 0.03$) (Supplementary Fig. 10). Moreover, the pull-down assay using an anti-HA tag antibody indicated that the interaction occurred only when both were fused with the Y15 peptide (Fig. 3e, lane 3). These interactions were abolished when Y15 in either protein was replaced with Y15(K9P) or when the tag was truncated.

We observed that most of the Y15-sfGFP granules were localized in the nucleus but not in the nucleolus (Supplementary Fig. 11). To investigate whether nuclear localization is necessary for assembly, we tested the assembly of Y15-sfGFP-NES (nuclear export sequence) in the cytosolic space. Y15-sfGFP-NES integrated into single micrometer-sized fluorescent granules in each cell (Fig. 3f), whereas cells expressing Y15-sfGFP-NES at low levels exhibited small puncta (Supplementary Fig. 11). Fluorescence recovery after photobleaching (FRAP) experiments revealed

that the granules are gel-like aggregates without fluidity, which is distinct from that of LLPS (Supplementary Fig. 11). Y15-sfGFP self-associates on the inner leaflet of the plasma membrane by incorporating a membrane-targeting sequence (CAAX)[27] from K-Ras. Confocal images showed that the green-condensed domains partially segregated on the membrane (Fig. 3g), in addition to vesicular localization probably caused by internalization. Interestingly, this condensed domain excluded the membrane marker protein (mCherry-CAAX) to some extent, suggesting that the phase has high levels of Y15-sfGFP-CAAX (Supplementary Fig. 11).

Next, we investigated the impact of the Y15-tagged site of proteins on clustering. In this study, three constructs—Y15-sfGFP-HA (tagged at the N-terminus), sfGFP-Y15-HA, and sfGFP-Y15 (tagged at the C-terminus)—were assessed by fluorescence anisotropy and fluorescence microscopy. Although sfGFP-Y15 showed a low propensity for assembly, the others were indistinct (Supplementary Fig. 12). Thus, for effective assembly, Y15 can be tagged at either the N-terminal or an internal site of the protein, but not at the C-terminal.

**Engineering of Y15-based highly ordered structure in cells.** Multivalent interactions between macromolecules can lead to the formation of highly ordered structures[28–30]. For example, the conjugation of the homo-oligomeric PB1 (Phox and Bem1) domain and tetrameric AG fluorescent protein produces cytoplasmic puncta[31]. Here, a single component of Y15-sfGFP formed puncta in living cells. However, this result is considered to be a combination of sfGFP dimerization and Y15-Y15 fiber formation, as sfGFP has been reported to form dimers at high concentrations[32]. To investigate the contribution of protein quaternary structure to granule formation, we assessed a Y15-fused monomeric AG, Y15-mAG, and tetrameric Y15-AG (Fig. 4a and b). Y15-AG drastically coalesced into puncta, and $89 \pm 4\%$ of the total protein was insoluble (Supplementary Fig. 13). In contrast, Y15-mAG was evenly dispersed throughout the cell, the majority ($63 \pm 6\%$) of which was soluble. Fluorescence anisotropy of mAG, however, decreased by Y15-tagging to the same degree as that of Y15-sfGFP, suggesting cluster formation (Fig. 4c). Thus, the Y15-Y15 interaction attributed to cluster formation but was insufficient to form micrometer-sized structures. We suspect that the additional protein–protein interaction cross-links the Y15-based fibrils to form highly ordered structures.

We investigated the characteristics of Y15-AG assemblies because Y15-AG self-association showed high potential as a scaffold for engineering micrometer-sized protein assemblies. Correlative electron microscopy of COS-7 cells expressing Y15-AG confirmed that the fluorescent granules were 100-nm-scaled structures without membrane separation (Fig. 4d and Supplementary Fig. 14). In the FRAP experiments, negligible fluorescent recovery was observed in the Y15-AG puncta, indicating that Y15-AG formed irreversible and rigid assemblies (Supplementary Fig. 15). Subcellular localization of Y15-AG clusters was both in the nucleus and adjacent to the endoplasmic reticulum in the cytosol (Supplementary Fig. 15); cytosolic localization is preferable for application. Furthermore, Y15-AG expression had little impact on cell viability (Supplementary Fig. 16).

Y15-AG scaffolds can be decorated by co-assembly with Y15-tagged client proteins. Y15-mCherry, as a model client, was co-transfected with Y15-AG, resulting in co-localization with the Y15-AG scaffold (Pearson $R = 0.89 \pm 0.09$), with $42 \pm 19\%$ of the Y15-mCherry recruited (Fig. 4e). In contrast, the single transfection of Y15-mCherry showed few fluorescent puncta (Supplementary

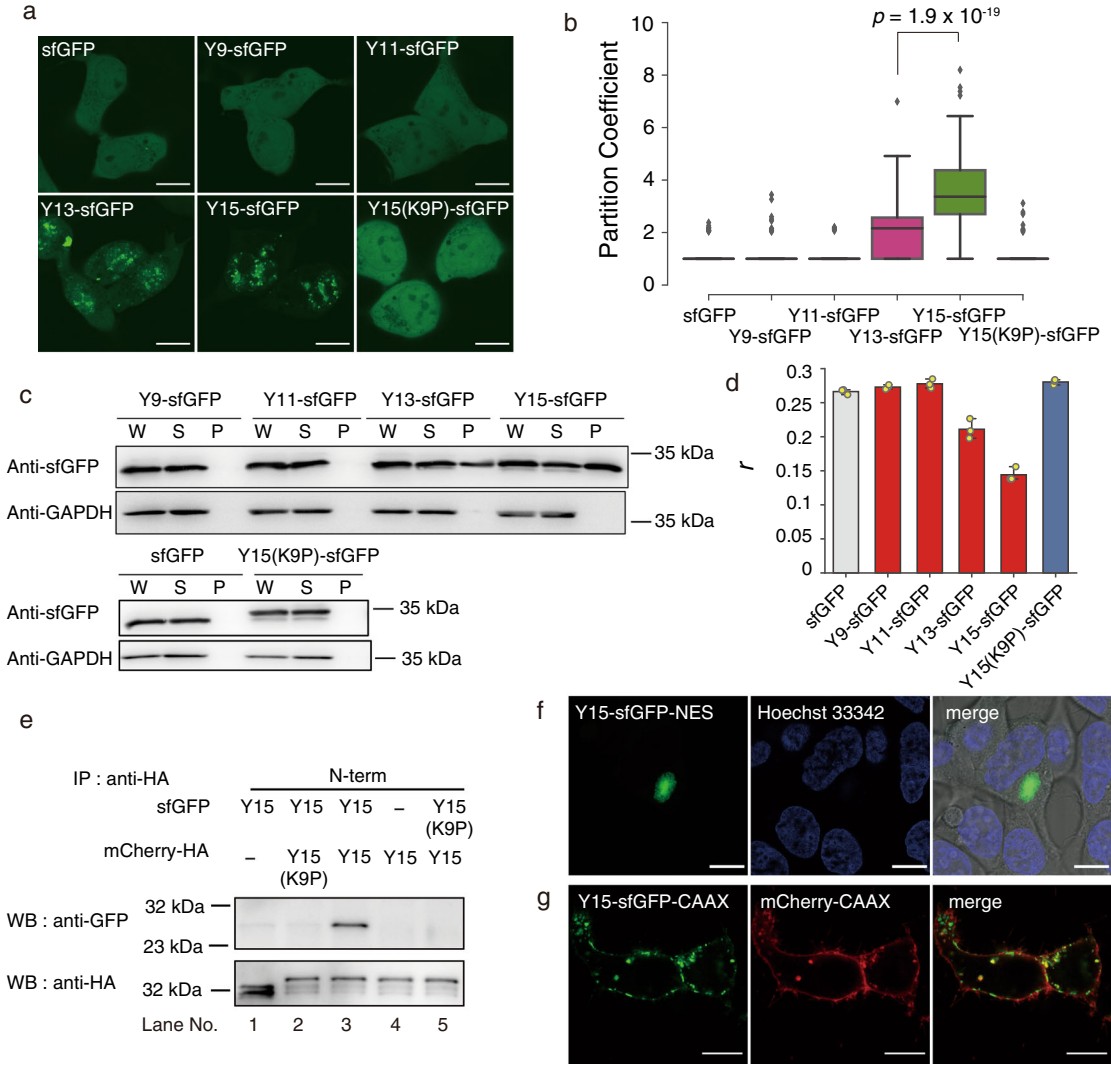

**Fig. 3 Self-assembly of Yn-fused sfGFPs in living cells. a** Fluorescence images of Yn-sfGFP-expressing HEK293 cells. HEK293 ($2 \times 10^5$) cells were transfected by lipofection of plasmids (500 ng), incubated for 2 days and then observed by CLSM. Scale bar, 10 μm. **b** Partition coefficient ($n = 100$ cells examined over three biologically independent experiments). The boxplots are presented with the elements: center line, median; box limits, Q1 and Q3; whiskers, 1.5× interquartile range; points, outliers. *p*-values; two-tailed paired *t*-tests. **c** Sedimentation assay of a cell lysate. The transfected cells were lysed in Ripa buffer. Whole lysates (W) were centrifuged to separate soluble (S) and pellet (P) fractions. These fractionated samples were analyzed by SDS-PAGE and western blotting using an anti-GFP antibody and an anti-GAPDH antibody (a marker for the soluble fraction). **d** Fluorescence anisotropy measurements of living cells ($n = 3$ biologically independent experiments). Transfected cells were cultured in a 96-well plate and fluorescence polarization was measured by a plate reader. Data are presented as mean values +/− SD. **e** Pull-down assay. Transfected HEK293 cells were lysed and immuno-precipitated using a mouse anti-HA tag antibody. To visualize sfGFP and mCherry-HA tag, a rabbit anti-sfGFP and a rabbit anti-HA antibody were used for western blotting. **f** Y15-sfGFP-NES self-assembly in the cytosolic space. Transfected HEK293 cells were stained by Hoechst 33342. Scale bar, 10 μm. **g** Y15-sfGFP-CAAX clusters on the inner leaflet of plasma membranes. HEK293 cells were co-transfected with Y15-sfGFP-CAAX and mCherry-CAAX (a marker for plasma membrane). Scale bar, 10 μm. Source data are provided as a Source data file.

Fig. 17), indicating that although the Y15-tagged monomeric proteins dispersed in cells, they can be integrated into the Y15-AG assembly with co-transfection.

**In-cell reconstitution of Y15-based Nck clusters induce actin polymerization.** We tested whether the Y15 system can function as a self-assembly platform for in-cell reconstitution of protein clusters. N-WASP signaling mediates local actin polymerization during phagocytosis and cell protrusions[33,34]. In this signaling pathway, the Nck-adaptor protein, consisting of one SH2 domain and three SH3 domains, locally interacts with phosphorylated receptors, such as nephrin. The Nck cluster recruits N-WASP by

multivalent interaction between the SH3 domains and proline-rich motifs, and the binding of the larger Arp2/3 complex with actin promotes filament nucleation. This mechanism has been studied by in vitro reconstitution[35,36], where purified Nck is immobilized on a glass surface or lipid bilayer and mixed with other components. The clustering of Nck SH3 domains and subsequent actin polymerization has also been demonstrated in living cells by using antibody-based systems[37] and CRY2oligo optogenetic clustering tools[38]. The Nck/N-WASP/Arp2/3 complex model was proposed by comparison of computational simulations with experimental results[39]. Moreover, recent studies[35,40] have revealed that phase separation driven by mul-tivalent interactions between Nck and N-WASP is the mechanism

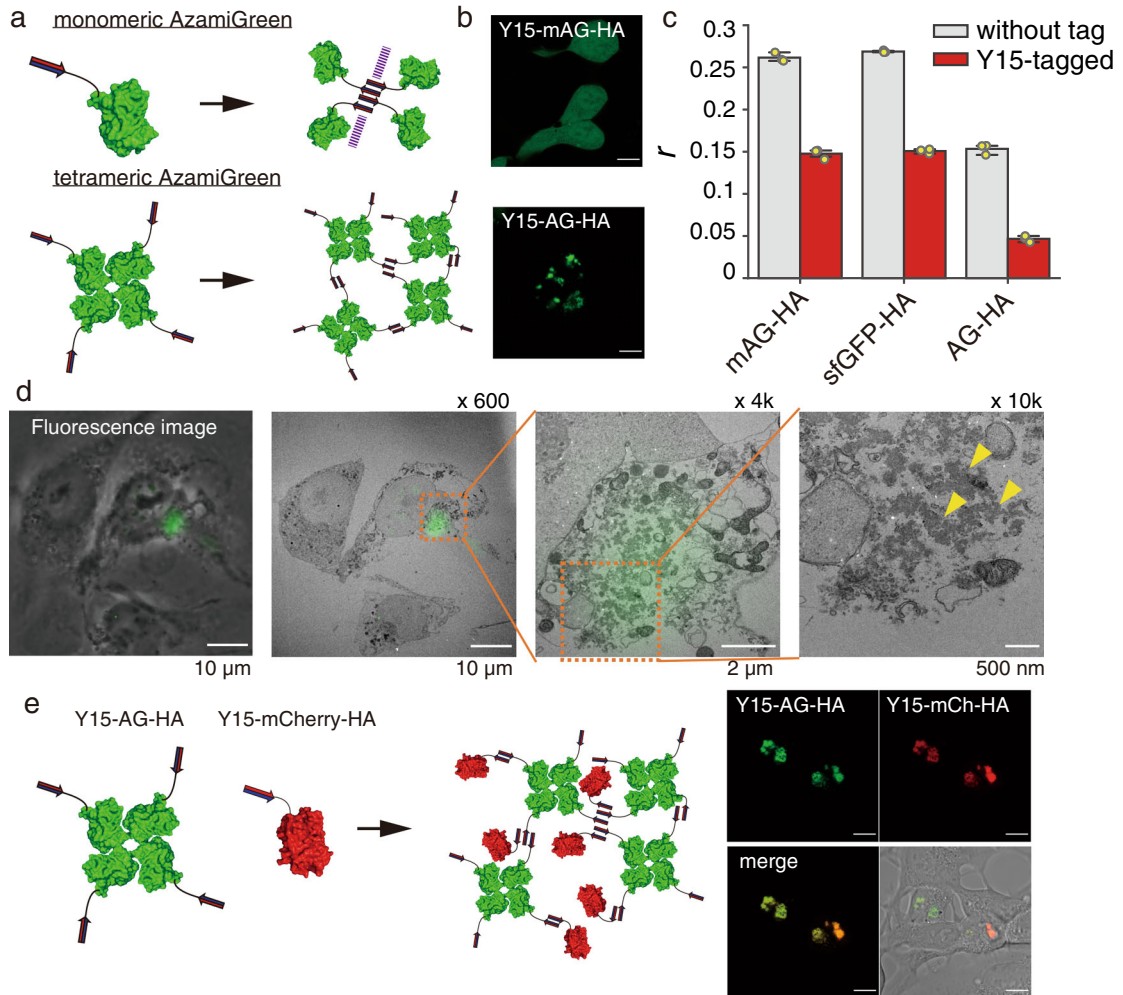

**Fig. 4 Engineering of protein aggregates by combining Y15 self-assembly and protein–protein interactions. a** Schematic illustration of the assembly of Y15-tagged monomeric AG and tetrameric AG. **b** Fluorescence images of HEK293 cells expressing Y15-mAG-HA and tetrameric Y15-AG-HA. Scale bar, 10 μm. **c** Fluorescence anisotropy measurements of transfected HEK293 cells ($n = 3$ biologically independent experiments). Data are presented as mean values $+/-$ SD. Source data are provided as a Source data file. **d** Correlative light and electron microscopy of Y15-AG expressing COS-7 cells, fluorescence image (left panel), overlay of the images (2nd and 3rd panels) and a higher magnification TEM image (right panel). Granular structures were not surrounded by membranes (arrowheads). **e** Fluorescence observation of co-transfected cells with Y15-AG-HA and Y15-mCherry-HA. Scale bar, 10 μm.

responsible for polymerization, and the relative stoichiometry of the components regulates the N-WASP dwell time, which correlates with actin polymerization[35].

We evaluated whether the Y15-mediated clustering of the Nck SH3 domains (residues 1–258) induces endogenous actin polymerization. To construct Nck-incorporated assemblies, we co-transfected Y15-tagged mCherry-Nck (Y15-mCherry-Nck) and Y15-AG scaffold into COS-7 cells (Fig. 5a). The filamentous actin (F-actin) structures were visualized by mTagBFP2-lifeact-7[41]. The Nck-doped assemblies were observed to co-localize with or adjacent to the F-actin (Fig. 5b), and the co-localization of endogenous N-WASP with the assemblies validated N-WASP signaling (Supplementary Fig. 18). The untagged mCherry-Nck failed to form granules, indicating that clustering of Nck proteins is critical for the complex formation (Supplementary Fig. 19).

In vitro structure–function analysis showed that the linker between the first two SH3 domains of Nck promotes phase separation and allosterically activates N-WASP[36,40]. Increasing the SH3 domain valency has been shown to enhance the N-WASP dwell time and concomitant actin assembly[35]. To verify the contribution of the sequence and domain valency to actin polymerization in cellular experiments, we tested a Nck(109–258)

truncation mutant containing the latter two SH3 domains (SH3B and SH3C) and a tandemly repeated 2×Nck(1–258) construct (Fig. 5c). F-actin in the assemblies was quantified by phalloidin-iFluor633 staining (Supplementary Fig. 20). 2×Nck(1–258) showed higher F-actin values than single Nck(1–258), while the truncated Nck(109–258) failed to promote actin polymerization, which is in good agreement with the results reported by in vitro experiments.

Finally, we tested the Nck density-dependency on actin polymerization. One advantage of self-assembling peptides is compositional control of assembly. Because the assembling moiety is a single component, the monomer's ratio directly reflects the compositions of co-assemblies. The plasmid concentration correlates linearly with the proportion of Y15-mCherry-Nck(1–258) in Y15-AG co-assemblies (Supplementary Fig. 21), facilitating the density-dependency assay. The results exhibited a bell-shaped relationship between the Nck proportion and F-actin intensity (Fig. 5d and Supplementary Fig. 22). Similar results have been reported for in vitro experiments using purified full-length N-WASP. In this report, the high concentration of Nck attenuates N-WASP activation, suggesting that excess Nck binding to WASP impairs the promotion of nucleation[42]. These results, however,

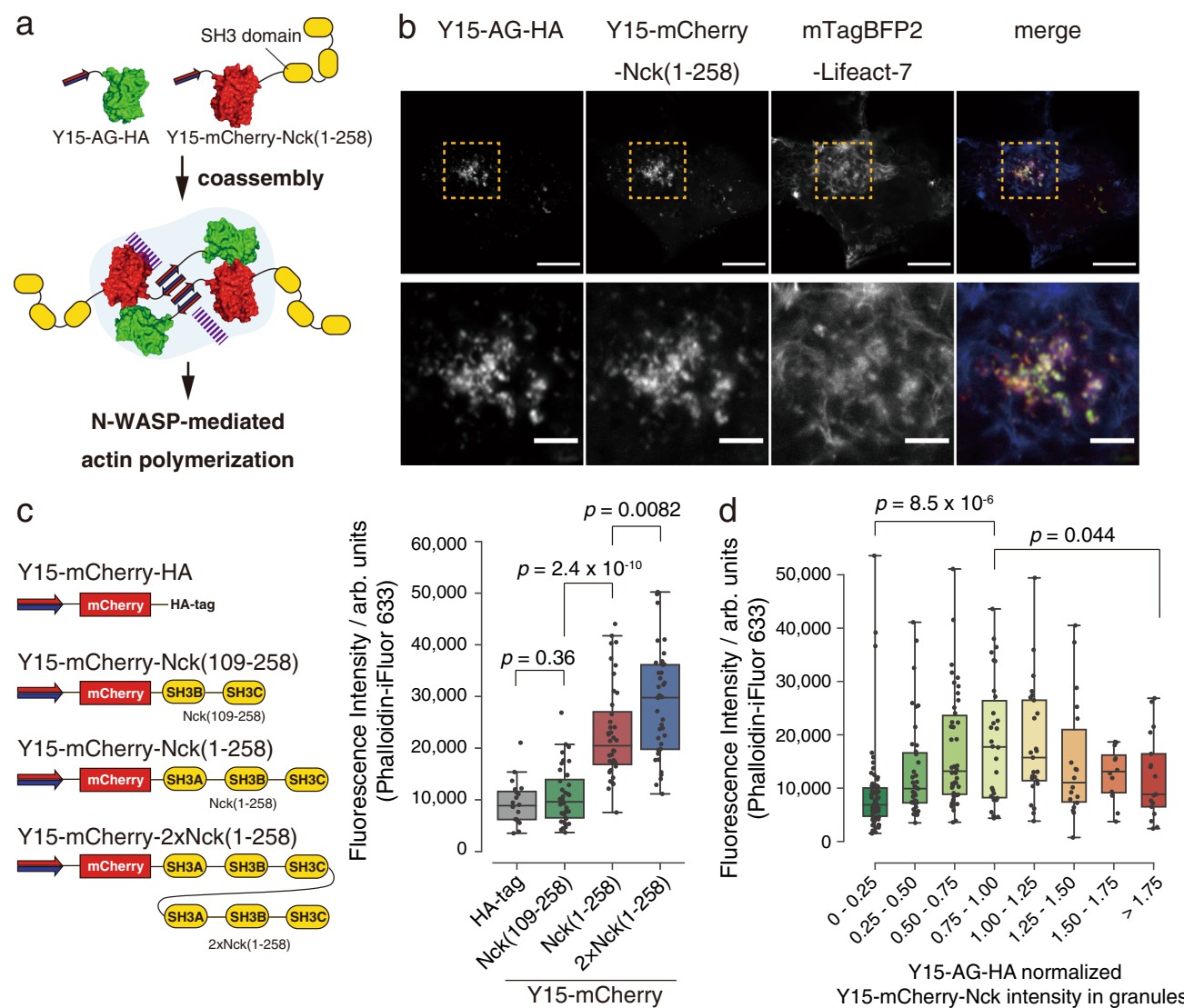

**Fig. 5 Intracellular reconstitution studies of Nck1 assemblies. a** Strategy for construction of Y15-based Nck assemblies in living cells. **b** CLSM images of engineered assemblies and polymerized actin in living COS-7 cells. COS-7 cells were co-transfected with the Y15-AG-HA-encoded plasmid (200 ng), Y15-mCherry-Nck(1–258)-encoded plasmid (200 ng) and mTagBFP2-Lifeact-7-encoded plasmid (100 ng). The cells were cultured for 2 days and CLSM observations were performed. The lower panels show enlarged images of ROIs given in the upper panels. Scale bar, 10 μm (upper) and 5 μm (lower). **c** Effect of Nck domains on actin polymerization (left panel, series of Y15-tagged Nck used in this study; right panel, fluorescence intensities of Phallodin-iFluor 633 in cytosolic Y15-based assemblies). Transfected cells were stained with Phallodin-iFluor 633 after fixation and permeabilization. The actin intensities in cytosolic granules were quantified ($n = 20, 40, 40,$ and $40$ cells examined over three biologically independent experiments). **d** Density-dependency of Nck(1–258) in assemblies on actin polymerization. COS-7 cells were transfected with mixtures of Y15-AG-HA-encoded plasmid (250 ng) and Y15-mCherry-Nck(1–258)-encoded plasmid (0, 62.5, 125, 187.5, or 250 ng) ($n = 50$ cells examined over three biologically independent experiments). The Nck densities were calculated as mCherry intensity normalized against AG intensity in Y15-based assemblies. The boxplots are presented with the elements: center line, median; box limits, Q1 and Q3; whiskers, 1.5× interquartile range; points, outliers. $p$-values; two-tailed paired $t$-tests. Source data are provided as a Source data file.

are inconsistent with reported data using antibody-induced aggregation methods, which showed that a high density of Nck is critical for actin polymerization on the membrane[39]. We hypothesize that the molecular mechanism responsible for actin polymerization differs between events on the membrane and those in the cytosol or solution because PIP2 (phosphatidylinositol 4,5-biphosphate) regulates N-WASP activation[43], and Nck is involved in the activation of N-WASP on PIP2-induced vesicles[44]. These results show that Y15-based assembly could be a platform for protein integration in living cells to facilitate in-cell reconstitution studies.

## Discussion
In this study, we developed a de novo designed self-assembling peptide tag, Y15, which shows the ability to form β-sheet nanofibers. Y15-tagged proteins integrate into fibril structures and can form clusters even in crude cellular contexts. Incorporation of Y15-tagged bioactive proteins can also functionalize assemblies.

Several de novo designed coiled-coil peptides have been used for both intracellular and extracellular applications. For example, a heterodimer of LK/LE peptides has been applied to control receptor activation[45] and the development of imaging tools[46]. In cells, orthogonal coiled-coil pairs have been shown to integrate

proteins[47] and regulate transcription[48]. Helical peptides forming cross-α structures produce intracellular protein condensates[49]. Compared with coiled-coil peptides, development of de novo β-sheet peptides is more difficult[50]. The individual coiled-coil peptide monomer forms a stable α-helical structure owing to complete backbone hydrogen bonding within the molecule. In contrast, β-sheet structures are generally stabilized by intermolecular interactions. Here, we successfully exploited a genetically encodable designed β-sheet tag that constructs supramolecular self-assemblies inside living mammalian cells by the multistep evaluation starting from the self-assembly of synthetic peptides.

In general, the recognition motifs used in synthetic biology are derived from natural domains or engineered domains. FK506 binding protein (FKBP) is a protein tag for chemically induced dimer/oligomer formation[51]. Various domains, such as SH3, can be used as scaffolds for integrating a subset of enzymes to accelerate cascade reactions[52]. Moreover, the concatenation of these domains generates artificial membraneless condensates by multivalent interactions in living cells[29,30]. Recently, optogenetic clustering tools, such as engineered CRY2, for spatiotemporal control of protein assembly have been reported[38,53]. Compared with such protein-based tags, the use of de novo SAPs has several advantages: First, SAPs (15 residues) are significantly smaller than protein-based recognition handles. For the formation of protein condensates using intrinsically disordered region (IDR) domains[54], a continuous repeat of IDRs (low-complexity (LC) domain) needs to be introduced, resulting in a large protein; second, usage of our SAP-based system is rational and straightforward. While the use of multiple components, such as heterodimerization of FKBP and FRB (FKBP-rapamycin binding) domain, requires strict regulation of their expression levels, SAP is essentially clustered within a single component. In addition, the clusters can be decorated with bioactive molecules by SAP-tagging, which is critical for optimizing biofunctionality; third, various SAPs have been reported in test tubes, and their sequence-structure relationships have been well characterized. For example, SAPs that responded to salt intensity, pH[55], enzyme[56], or metal ions[11], and SAPs with catalytic activity[57,58] have been reported. For future developments, cluster formation with stimuli-responsiveness and intracellular applications of functional SAPs can be expected.

The Y15-based assemblies were irreversible and showed solid-like properties. The hydrophobic interfaces between β-sheets of the de novo designed peptide are well defined and are generally dry. Although the LC domain of the fused in sarcoma (FUS) protein, a representative protein in LLPS, also forms a cross-β structure, the interface is hydrophilic and contains water molecules, unlike amyloid fibrils[59,60]. The wet interface appears to reduce the assembly stability, resulting in liquid-like properties. We expect that investigating of the relationship between SAP sequences and the physical properties of the assemblies will lead to successful mimicry and artificial constitution of LLPS in cells. We also envisage that Y15-based and Y15-related biomaterials will be used to elucidate the function of protein complexes in cells. As shown in the example of the integration of Nck, this system can be used as a platform for reconstituting natural protein complexes in cells to verify their detailed mechanisms.

## Methods

**Peptide synthesis.** Peptides were synthesized by standard Fmoc solid-phase synthesis. The first Fmoc-amino acid was loaded on Wang resin (Sigma-Aldrich) by esterification with $N,N'$-Dicyclohexylcarbodiimide/4-dimethylaminopyridine. The peptide chains were elongated by using Fmoc-amino acids (3 eq.), $N,N$-diisopropylethylamine (DIPEA, 6 eq.), HBTU (3 eq.), and HOBt (3 eq.) in

$N$-methylpyrrolidone (NMP) for coupling and 20% piperidine in NMP for Fmoc deprotection. All peptides were cleaved from the resin and deprotected by treating with trifluoroacetic acid (TFA)/triisopropylsilane/water (95:2.5:2.5 v/v). Crude peptides were precipitated in cold diethyl ether and purified by semi-preparative RP-HPLC running a linear gradient of acetonitrile at a flow rate of 3.0 mL min$^{-1}$ on a COSMOSIL 5C18-AR-300 or 5C18-AR-II packed column (10 × 250 mm) with ultraviolet detection at 220 nm. We used eluent A (0.1% TFA ultra-pure water) and eluent B (0.08% TFA acetonitrile). Pure fractions were analyzed by analytical RP-HPLC and MALDI-TOF MS. The pure peptides were lyophilized.

**Peptide assembly assay.** Peptide powders were dissolved in ultra-pure water or DMSO to prepare peptide stocks. The stock concentration was determined by Tyr absorbance. The stock peptides were diluted with ultra-pure water and 10× PBS or 10× PB to a concentration of 5–50 μM. The testing solution was incubated at 37 °C for 1 day. Thioflavin-T (final 25 μM) was added to the peptide solution (5–50 μM) for the thioflavin-T assay, and fluorescence spectra were measured by an F-7000 (Hitachi). For IR measurements, peptide stocks were diluted with ultra-pure water and 10× PBS to a concentration of 400 μM and incubated at 37 °C for 18 h. Ten microliters of the peptide solution was placed on the reflective surface of the FT-IR instrument (IRPrestige-21 with DuraSampl IR-II, Shimadzu). ATR-FTIR (Attenuated total reflection-Fourier transform infrared) spectra of the dried peptide film were acquired.

**CD measurements.** Ten or 50 μM peptide solutions were prepared in the same manner as the thioflavin-T assay and examined by CD spectroscopy. CD spectra were recorded using a J-1100 (JASCO) spectropolarimeter with spectra manager (2.8.0.4) and a quartz cell with a 1.0 mm pathlength at 25 °C. Millidegrees were converted to mean residue ellipticity.

**TEM observations.** Peptide or protein solutions were placed on collodion-coated copper EM grids. Excess solutions were removed with filter paper, and the coated side on the grids was washed with filtered ultra-pure water. The samples were negatively stained by Nano-W®, followed by washing with filtered ultra-pure water and dried. All images were obtained by using a JEOL 1400Plus electron microscope.

**Construction of plasmids.** The *sfGFP* was inserted into the pCI-neo vector (Promega, E1841) at *Sal*I-*Not*I (for construction of Yn-sfGFP) or *Eco*RI-*Sal*I sites (for construction of Y15-sfGFP-HA/NES/CAAX) by using a standard restriction enzyme cloning technique. For Yn-sfGFP constructs, the oligo DNAs corresponding to the Yn peptide were ordered (FASMAC), phosphorylated by T4 polynucleotide kinase (PNK, Takara, 2021 S) and annealed overnight (inactivation of PNK: 70 °C for 20 min; denaturing: 90 °C for 5 min; annealing: cooling at −1 °C/min). The annealed DNA was inserted into the pCI-sfGFP vector at *Nhe*I-*Sal*I sites (Supplementary Table 1). For the construction of Y15-sfGFP-HA, the Y15 and HA sequences prepared from ordered oligo DNA were inserted into pCI-sfGFP at *Nhe*I-*Eco*RI and *Sal*I-*Not*I sites, respectively (Supplementary Table 2). Y15-sfGFP-NES and Y15-sfGFP-CAAX were constructed using the same procedures. Y15-mCherry-HA, Y15-mAG-HA, and Y15-AG-HA plasmids were generated by substitution of *sfGFP*. *mAG* and *AG* were purchased from MBL and *Nck1* was cloned by nested-PCR from Human Brain, whole Marathon®-Ready cDNA (Clonetech, 639301). The gene of residues 1–258 of Nck1 (SH3 domains) was subcloned by PCR and inserted into pCI-Y15-mCherry at *Sal*I-*Not*I sites to produce Y15-mCherry-Nck (Supplementary Table 3). For expression of Yn-sfGFP in bacterial cells, genes of Yn-sfGFP were cloned into pET vectors between *Nhe*I and *Not*I sites.

**Protein purification of Yn-sfGFP.** Recombinant plasmids were transformed into BL21(DE3) Codon Plus RIPL *E. coli* cells (Agilent). Bacteria were grown in LB medium at 37 °C until the OD$_{600}$ reached 1.0. Protein overexpression was then induced by adding isopropyl β-D-1-thiogalactopyranoside to a final concentration of 1 mM, and cells were culture for a further 5 h at 37 °C and 200 rpm. After centrifugation, the cell pellet was suspended in lysis buffer (20 mM Tris, 150 mM NaCl, 1% Triton-X, 1 mM PMSF, pH 8.0), and cells lysed by sonication. The majority of Yn-sfGFP was expressed in inclusion bodies. The inclusion was resuspended in urea buffer (8 M urea, 50 mM sodium phosphate, 300 mM NaCl, pH 7.2) and clarified by centrifugation. The supernatant was subjected to TALON® resin. After binding at room temperature, the column was washed with 10 column volumes of urea buffer. The protein was eluted with elution buffer (8 M urea, 45 mM sodium phosphate, 270 mM NaCl, 150 mM imidazole, pH 7.2). The eluate was dialyzed overnight against a low salt buffer (1 mM EDTA, 10 mM sodium phosphate, pH 7.2) using dialysis tubing with a 12–14 kDa cut-off membrane (Spectra/Por®). The majority of recombinant sfGFP and Y15(K9P)-sfGFP existed in the soluble fraction after cell lysis. The soluble fractions were purified by TALON resin using a wash buffer (50 mM sodium phosphate, 300 mM NaCl, pH 7.2) and elution buffer (45 mM sodium phosphate, 270 mM NaCl, 150 mM imidazole, pH 7.2). The protein concentrations were determined by the BCA assay. Before assembly tests, all variants were denatured in 5 M urea buffer (5 M urea,

10 mM sodium phosphate, 1 mM EDTA, pH 7.2), then dialyzed against 1 M urea buffer (1 M urea, 10 mM sodium phosphate, 1 mM EDTA, pH 7.2, once) and sodium phosphate buffer (10 mM sodium phosphate, 1 mM EDTA, pH 7.2, twice).

**Cell culture and transfection**. HEK293 and COS-7 cells were provided by the RIKEN BRC through the National BioResource Project of the MEXT/AMED, Japan. Both cell types were cultured on 60-mm dishes in 5 mL Dulbecco's modified Eagle's medium (DMEM, Sigma) supplemented with 10% fetal bovine serum and penicillin-streptomycin at 37 °C with 5% $CO_2$ (MCO-5AC, PHC). HEK293 ($2 \times 10^5$ cells/well) or COS-7 ($1 \times 10^5$ cells/well) cells were seeded in 24-well plates for 6–16 h before transfection. The cells were transfected with 500 ng plasmids by lipofection (Lipofectamine® 3000, Thermo). The transfected cells were incubated for 1 day in a $CO_2$ incubator and analyzed. DsRed2-Peroxisomes-4 was a gift from Michael Davidson (Addgene plasmid #54503; http://n2t.net/addgene:54503; RRID: Addgene_54503). mCherry-ER-3 (Addgene plasmid #55041; http://n2t.net/addgene:55041; RRID:Addgene_55041) and mTagBFP2-Lifeact-7 (Addgene plasmid #54602; http://n2t.net/addgene:54602; RRID:Addgene_54602) were also a gift from Michael Davidson.

**Actin staining**. Transfected COS-7 cells were fixed with a 4% paraformaldehyde phosphate buffer (Nacalai Tesque) for 15 min at room temperature and permeabilized by 0.1% Triton X-100 in PBS for 10 min at room temperature. The fixed cells were stained by a Phalloidin-iFluor 633 working solution (Abcam, ab176758) for 1 h. After three washes in PBS, mounting media ProLong (Thermo Fisher) was added.

**Fluorescence microscopy**. Transfected cells were placed on a 35-mm glass-bottom dish (IWAKI) coated with poly-L-Lys. Before observation, the media was substituted with D-MEM (HEPES, no Phenol Red). Samples were observed by confocal laser fluorescence microscopy (CLSM; LSM780, Zeiss) with ZEN2.3 (black) software. We used a 405 nm laser diode for Hoechst 33342 (Dojindo) and DAPI, a 488 nm argon laser for sfGFP, mAG, and AG, a 561 nm diode-pumped solid-state laser for mCherry, DsRed2, and Nucleolus Bright Red (Dojindo), and a 633 nm He–Ne laser for Alexa fluor 633.

For imaging the proteins in PBS, samples were placed on a glass-bottom 96-well plate and observed by fluorescent microscopy (ECLIPSE TI-FL, Nikon or AxioObserverZ1, Zeiss, ZEN2.6 (blue) software).

**Image analysis**. Image analysis was performed in Fiji (ImageJ) and Microsoft Excel software. For partition coefficient calculations of Yn-sfGFPs, initially, cell regions were identified by percentile (0.8) thresholding on each fluorescence image. Regions showing more than twice the average value of the cell regions were identified as granules. The ratio values of the fluorescence intensity inside granules versus the bulk region were calculated for the partition coefficient. The partition coefficient value was set to 1 for cells without granules. For quantification of actin intensities in fluorescent granules, we performed the following processes. Initially, the area of the nucleus of transfected cells was eliminated, and the area of AzamiGreen fluorescent granules was determined by Otsu's threshold (ImageJ). The fluorescence intensities within the region for all three channels were then calculated.

**Correlative electron microscopy**. COS-7 cells were seeded one day after transfection onto carbon-coated glass-bottom dishes with a grid pattern. The cells were incubated for 1 day and prefixed with 4% PFA-0.1% GA (glutaraldehyde, EM. S. #3052-1) for 15 min at room temperature. The cells were observed immediately by using a fluorescent microscope (Nikon ECLIPSE TI-FL). After observation, the cells were fixed with 2.5% GA in 0.1 M sodium phosphate buffer (pH 7.4) for 30 min. Samples were rinsed three times with 0.1 M sodium phosphate buffer on ice and fixed with 1.0% $OsO_4$ (EM. S. #3020-3) in 0.1 M sodium phosphate buffer (pH 7.4) for 30 min. After rinsing with ice-cold buffer, samples were dehydrated by substitution with 50% EtOH, 70% EtOH, 90% EtOH, and 100% EtOH (three times). Epoxy resins (mixture of MNA (TAAB #3471), EPON812 (TAAB #342), DDSA (TAAB, #3461), and DMP-30 (TAAB, #3481)) were used to embed specimens for 48 h at 60 °C. The regions observed by fluorescence microscopy were trimmed from the resin block and sectioned (60–70 nm) by using an ultramicrotome Leica UC-7. The sections were stained with EM stainer (EM. S #336) for 60 min or 4% samarium chloride for 20 min, followed by Reynold's stain for 5 min. Images were recorded by using a JEOL 1400Plus electron microscope operating at an accelerating voltage of 80 kV with TEM CENTER software.

**Cell lysis and fractionation**. Transfected cells in a 24-well plate were lysed on ice using Ripa buffer (Nacalai) containing a 1% protease inhibitor cocktail (Nacalai). The lysed cell samples were centrifuged ($16,000 \times g$, 10 min). The supernatants were collected as soluble fractions. The pellets were washed with Ripa buffer, centrifuged and the supernatant removed to yield the insoluble fractions. Both fractions were diluted or dissolved with 2× Laemmli buffer (125 mM Tris-HCl, 20% glycerol, 4% SDS, 0.01% bromophenol blue, 100 mM dithiothreitol, pH 6.8)

and incubated at 95 °C for 10 min. The samples were loaded onto SDS-PAGE (12.5%) and then electro-transferred onto an Immobilon PVDF membrane (Millipore). After blocking with 0.5% skimmed milk in TBS-T, the membrane was incubated overnight at 4 °C with a rabbit anti-GFP antibody (GeneTex, GTX113617, 1:2500 dilution), a rabbit anti-HA-tag antibody (MBL, 561, 1:2500 dilution), or a mouse anti-GAPDH antibody (MBL, M171-3, 1:2500 dilution). After three washes with TBS-T, the membrane was incubated with a goat anti-rabbit IgG-HRP conjugate (Abcam, ab6721, 1:2500 dilution) or goat anti-mouse IgG-HPR conjugate (Promega, W402B, 1:2500 dilution) solution in TBS-T. The proteins were detected by chemiluminescence analysis using ECL prime (RPN2232) and a WSE-6100 LuminoGraph I instrument (ATTO) with ImageSaver6.

**Pull-down assay**. Transfected cells were lysed in Ripa buffer (1% protease inhibitor cocktail) after washing with PBS. SureBeads Protein G Magnetic Beads (BioRad, 161-4021) treated with a mouse anti-HA antibody (MBL, M180-3S, 1:300 dilution) were added to the lysate and incubated overnight at 4 °C. The beads were rinsed three times with PBS-T buffer. Proteins bound to the beads were eluted with 1× Laemmli buffer by boiling at 95 °C for 5 min. These samples were used in SDS-PAGE and western blotting analysis.

**Fluorescence anisotropy measurements**. For the in vitro assay, 5 μM Yn-sfGFP in sodium phosphate buffer (pH 7.2) with 1 mM EDTA was incubated for 24 h at 37 °C. The fluorescence intensity and anisotropy were then measured by using an ARVO MX plate reader (Perkin Elmer) with Wallac1420 Workstation. For cell experiments, transfected HEK293 cells were incubated in a 96-well plate for 1 day. The media was substituted with D-MEM (HEPES, no Phenol Red). The fluorescent anisotropy was measured by the ARVO MX plate reader (Perkin Elmer). Both vertical and horizontal fluorescence from transfected cells were subtracted by those from non-transfected HEK293 cells to reduce the effect of autofluorescence from HEK293 cells. Fluorescent anisotropy values were calculated using the background-subtracted intensities.

**Statistics and reproducibility**. All assembly tests, including cell experiments, were biologically replicated at least three times independently. All attempts at replication were successful. The n values used for statistical analysis are shown in the figure legends. All bar graphs show mean ± SD of independent experiments. Unpaired two-tailed Student's t-tests were used for comparing two groups. All boxplots are presented with the elements: center line, median; box limits, Q1 and Q3; whiskers, 1.5× interquartile range; points, outliers. The exact p-values are given in the figure legends, and a p-value <0.05 was defined as statistically significant.

**Reporting summary**. Further information on research design is available in the Nature Research Reporting Summary linked to this article.

## Data availability
The source data underlying all figures, including supplementary figures, are provided with this paper. All data are available from the corresponding author upon reasonable request. Source data are provided with this paper.

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

## Acknowledgements

We thank Open Research Facilities for Life Science and Technology, Tokyo Institute of Technology for equipment and technical support. We thank the Biomaterials Analysis Division, Tokyo Institute of Technology for DNA sequencing and for technical assistance of TEM observation. We also thank Ms. Keiko Ikeda for preparing and observing samples with correlative electron microscopy. This work was supported by the JSPS KAKENHI Grant (21K14739), the JGC-S Scholarship Foundation, and a Grant-in-Aid for Challenging Research, Organization of Fundamental Research, Tokyo Institute of Technology. We thank Martin Cheung PhD, from Edanz Group (https://en-author-services.edanz.com/ac) for editing a draft of this manuscript.

## Author contributions

T.M., H.T., and H.M. designed the project. T.N. synthesized peptides and perform assembly test. T.M., T.N., and M.H. constructed the expression plasmids. T.M. and T.N. expressed and purified proteins and test their assembling properties in test tube. T.M., T.N., K.K., and M.H. perform cell experiments. The manuscript was written by T.M. and H.M., according to discussion with all authors.

## Competing interests

The authors declare no competing interests.
