## [Peer Review File · Nature Communications]

REVIEWER COMMENTS

Reviewer #1 (Remarks to the Author):

Miki et al. report several peptide-protein conjugates that are capable of functionally assemble in living cells. The field of self-assembling peptides has exploded in recent years with a number of functional applications reported. However the examples of biocompatible applications were far in between (and none as nice as this one), so I'm enthusiastically supporting publication of this manuscript in Nat Commun. This work shows that rationally designed interactions can aid (and potentially drive) biological function with lots of interesting future possibilities.

The peptide part of the work is well done and convincing and I have no issues with it. I'm less qualified to judge the microscopy part.

Few minor points I'd like to be addressed prior to publication.

Nck's role (and what it is) is not properly described in the abstract. Also the abstract doesn't do the paper justice. The authors should consider rewriting it. The introduction was also a bit jumpy, some polishing might be needed.

Which linkers if any are used to fuse the peptides to the proteins? It's be actually nice to see complete sequences of the constructs in the SI.

Along the same lines the graphical schemes show peptide dimers, trimers, tetramers, etc. It's probably very unlikely as peptides tend to self-assemble into long fibrils. I understand that the authors had to do it for clarity, but it might be prudent to

Reviewer #2 (Remarks to the Author):

Manuscript by Miki et al. describe the design and characterization of self-assembling peptide that induces formation of oligomers and brings the selected genetically fused domains into the proximity.

The approach is very similar to the reports by Collier group (e.g. Hudalla et al, Q11 peptide, KFE8 etc), in selection of an amphipathic beta-sheet propensity peptide containing high fraction of aromatic residues. Also attachment of the cargo fluorescent protein is similar.

In this report several versions of the peptide of different length are evaluated and the minimal size length is identified. Fusion with fluorescent proteins or Nck1 SH3 domains trigger formation of assemblies within mammalian cells as expected.

Several techniques are used to characterize the assemblies, including electron microscopy, light scattering techniques, fluorescence microscopy that suggest formation of fibrils.

It is not clear in which ways this peptide scaffold differs from other previously reported self-assembling peptides and why this would find other use beyond previously described applications such as e.g. for vaccines, materials, scaffolds with increased catalytic efficiency or guest protein assemblies.

The main novelty is demonstration of selfassembly inside mammalian cells. This has been used to demonstrate the effect of Nck clusters on actin polymerization. This is not a novel finding as the stoichiometry of the assembly had been determined previously but might provide a useful tool, particularly if this process could be regulated externally, which has not been done before.

An interesting aspect of the report is on formation of condensates. The authors conclude that Y15 should not be positioned at the C-terminus although it is unexpected that the addition of an HA tag at the C-terminus evades this problem since the HA tag is very small. Formation of condensates is mentioned with references to liquid phase separation, however FRAP results suggest that they likely form immobile aggregates, therefore the term condensates may not be most appropriate.

I don't think this contribution stringly contributes to our understanding or application of supramolecular assemblies in cells.

Reviewer #3 (Remarks to the Author):

In the manuscript by Miki and colleagues, the authors investigate the use of short peptides that contain alternating hydrophobic and charged residues creating function cellular compartments. They find that a short, 15 residue peptide (Y15) readily self-assembles into structures both in vitro and in cells. The authors then demonstrate that the structures formed by their peptide can be functionalized. Using an Nck signaling pathways to the actin cytoskeleton, the authors show that these structures are able to promote local actin polymerization.

This study provides a potentially valuable tool for manipulating cellular function using self-assembling platforms linked to specific signaling pathways or other cellular functions. However, this reviewer found that the manuscript was a bit confusing because of the invocation of phase separation and then stating that the structures aren't phase separated. This confusion can be cleared up by performing the proper experiments that can specifically show whether these structures are phase separated or not; these are described below. Importantly, understanding the biophysics of structure formation is essential for promoting this experimental platform, regardless of whether these structures form through phase separation or another mechanism like polymerization. This author also found the description of actin polymerization by Nck and the dissection of the signaling pathway to be lacking; suggestions are also included below. However, because of the novelty of this particular self-assembling platform and its potential usefulness to a broad range of readers, if the authors are able to address this reviewer's concerns, this manuscript will be suitable for publication in Nature Communications.

Major Comments:

1) The authors switch between phase separation and non-phase separated structures throughout the manuscript. In the introduction, they refer to phase separation, yet their results may indicate that the Y15 structures are not phase separated, leaving this review confused as to why phase separation was invoked early on in the paper as the mechanism pointed to for self-assembly. Depending on the results of the experiments in Major Comments 2, draft the manuscript to focus on phase separation or self-assembly into an oligomer, not both.

2) The authors do not definitely show the mechanism of self-assembly and should perform experiments to clarify either phase separation or oligomer formation. FRAP of cellular condensates isn't indicative of the mechanism of formation, rather of the dynamics of molecules. Some structures initially undergo liquid-liquid phase separation and then mature into gel-like condensates (Lin et al. Mol Cell 2015) and would have similar dynamics as what is observed in the cellular experiments here. Because valency is important (smaller peptides do not self-assemble and increasing the valency of AG from monomer to tetramer increases the propensity to form cellular structures), phase separation is a possible mechanism that underlies the formation of these structures. The authors should perform phase behavior experiments in which they 1) increase the concentration of Y15 peptide in their buffer of choice to determine if there is a critical concentration above which structures form and 2) increase salt concentration to determine if the structure assembly is regulated by salt concentration or increase / decrease the temperature at which their assays are performed to determine if there is a temperature dependence for structure assembly. Because their peptide forms structures through a combination of hydrophobic interactions, pi-pi interactions, and electrostatic interactions, changing the salt concentration would be informative; at low salt, pi-pi and electrostatic interactions would drive phase separation and at high salt, hydrophobic interactions would drive phase separation.

3) The experiments in which the authors investigate a requirement for nuclear localization shouldn't involve the membrane. If these structures form through phase separation, targeting them to a membrane, will change the dimensionality of system and the phase behavior. These experiments

should be repeated using a nuclear export sequence to target the peptides to the cytoplasm, not the plasma membrane.

4) Nck-induced actin polymerization needs to be clarified. Nck-dependent actin polymerization has been studied extensively in cells (Rivera et al, Curr Biol 2004, Ditlev et al J Cell Biol 2012, Taslimi et al Nat Comms 2014). It is the phase separation of Nephrin, Nck, N-WASP that controls the stoichiometry resulting in optimal actin polymerization (Case et al Science 2019 provides a lens into the mechanism that regulates the results in Ditlev et al 2012 and Su et al Science 2016), not just oligomerization. Because phase separation is the mechanism by which Nck promotes actin polymerization through N-WASP and the Arp2/3 complex, the experiments in which the first SH3 domain and Linker are deleted (both of which contribute to phase separation (Banjade et al PNAS 2015) and the double Nck support phase separation as the mechanism driving local actin polymerization. In these scenarios, the authors alter the valency of Nck and thus likely the propensity of phase separate. The authors also state that by varying the amount of Nck in condensates, actin polymerization changes. This isn't entirely accurate, as Ditlev et al J Cell Biol 2012 clearly showed that the amount of Nck in clusters isn't the driver of actin polymerization, density and stoichiometry with N-WASP and Arp2/3 complex are the driving factors. No additional experiments are needed. Rather, the authors experiments should be considered in light of these previous studies and this section should be rewritten.

Minor Comments:

1) Line 4, 'Microtubes' should be microtubules.

2) Line 5, 'Biomolecular condensates, or protein phase separation' aren't analogous terms. Phase separation can drive biomolecular condensate formation. This should be fixed.

3) Figure 1D should include representative EM images of each peptide at the condition shown for Y15, if the images are available.

4) It appears that the structure of the Y15 and Y15-sfGFP are different. Could the authors comment on why? Does the sfGFP alter the structural layout or might the Y15 interact with the surface of sfGFP to induce different structures?

5) The method used to quantify the Y15 content of the cellular structures may not be the best or easiest way to accomplish this. The authors should repeat analysis of the images and measure partition coefficient (Fluorescence inside structures vs. outside structure) to quantify the differences between Y15, Y13, and other Y peptides as applicable (See Banani et al Cell 2016 for partition coefficient information).

Response to Reviewer's comments
Response to Reviewer 1's comments

Comment

Miki et al. report several peptide-protein conjugates that are capable of functionally assemble in living cells. The field of self-assembling peptides has exploded in recent years with a number of functional applications reported. However the examples of biocompatible applications were far in between (and none as nice as this one), so I'm enthusiastically supporting publication of this manuscript in Nat Commun. This work shows that rationally designed interactions can aid (and potentially drive) biological function with lots of interesting future possibilities.

The peptide part of the work is well done and convincing and I have no issues with it. I'm less qualified to judge the microscopy part.

Few minor points I'd like to be addressed prior to publication.

Nck's role (and what it is) is not properly described in the abstract. Also the abstract doesn't do the paper justice. The authors should consider rewriting it. The introduction was also a bit jumpy, some polishing might be needed.

Which linkers if any are used to fuse the peptides to the proteins? It's be actually nice to see complete sequences of the constructs in the SI.

Along the same lines the graphical schemes show peptide dimers, trimers, tetramers, etc. It's probably very unlikely as peptides tend to self-assemble into long fibrils. I understand that the authors had to do it for clarity, but it might be prudent to

Our response:

Thank you for the important comments and advice. We carefully take the comments into consideration on amending our manuscript as shown in below. All the revision we made are highlighted in yellow in the revised manuscript.

Comment 1

Nck's role (and what it is) is not properly described in the abstract. Also the abstract doesn't do the paper justice. The authors should consider rewriting it.

Our response:

We rewrote the abstract as shown in the following sentence. We simplified our logic and added description about Nck and the role.

Modification in the maintext: (modified sites were highlighted)

“De novo designed self-assembling peptides (SAPs) are promising building blocks of supramolecular biomaterials, which can fulfill a wide range of applications, such as scaffolds for tissue culture, three-dimensional cell culture, and vaccine adjuvants. Nevertheless, the use of SAPs in intracellular spaces has mostly been unexplored. Here, we report a self-assembling peptide, Y15 (YEYKYEYKYEYKYEY), which readily forms β -sheet structures to facilitate bottom-up synthesis of functional protein assemblies in living cells. Superfolder green fluorescent protein (sfGFP) fused to Y15 assembled into fibrils and was observed as fluorescent puncta in mammalian cells. Y15 self-assembly was validated by fluorescence anisotropy and pull-down assays. By using the Y15 platform, we demonstrated intracellular reconstitution of Nck assembly, a Src-homology 2 and 3 domain-containing adaptor protein. The artificial clusters of Nck induced N-WASP (neural Wiskott-Aldrich syndrome protein)-mediated actin polymerization, and the functional importance of Nck domain valency and density was evaluated.”

Comment 2

The introduction was also a bit jumpy, some polishing might be needed.

Our response:

We clarified our purpose and strategies. Moreover, as reviewers 2 and 3 pointed out, the Y15 system does not show liquid-liquid phase separation, so the description of the condensates in introduction has been deleted.

Modification in the maintext: (modified sites were highlighted)

“In natural biological systems, non-covalent interactions drive macromolecules to spontaneously organize into supramolecular structures, giving rise to complex functionality. For example, cytoskeleton components, such as actin filaments and microtubules, which play central roles in cell shaping, migration, and intracellular transportation, form in this manner¹. Therefore, platform techniques that build artificial protein assemblies in living systems are crucial for achieving the ultimate goal of synthetic biology: elucidating biological systems through mimicry and

generating novel systems with artificial functions^{2,3}.

Various de novo designed peptides have been developed to produce valuable and practical supramolecular biomaterials in bottom-up strategies. In this approach, ...”

Comment 3

Which linkers if any are used to fuse the peptides to the proteins? It's be actually nice to see complete sequences of the constructs in the SI.

Our response:

We added all complete protein sequences in the SI (Table S4 and S5).

Comment 4

Along the same lines the graphical schemes show peptide dimers, trimers, tetramers, etc. It's probably very unlikely as peptides tend to self-assemble into long fibrils. I understand that the authors had to do it for clarity, but it might be prudent to

Our response:

As the reviewer comment's, Y15 peptides self-assemble into long fibrils. We added dashed lines to indicate the polymeric assemblies in graphical illustrations. Although we are not sure whether Y15 tag form oligomer or longer fibrils in the case of Y15-tetrameric AG, we kept the original style.

Modification in the maintext:

Figure 2a: We added dashed lines in schematic illustration.

Figure 4a: We added dashed lines in schematic illustration.

Figure 5a: We added dashed lines in schematic illustration.

Response to Reviewer 2's comments

Comments

Manuscript by Miki et al. describe the design and characterization of self-assembling peptide that induces formation of oligomers and brings the selected genetically fused domains into the proximity.

The approach is very similar to the reports by Collier group (e.g. Hudalla et al, Q11 peptide, KFE8 etc), in selection of an amphipathic beta-sheet propensity peptide containing high fraction of aromatic residues. Also attachment of the cargo fluorescent protein is similar.

In this report several versions of the peptide of different length are evaluated and the minimal size length is identified. Fusion with fluorescent proteins or Nck1 SH3 domains trigger formation of assemblies within mammalian cells as expected.

Several techniques are used to characterize the assemblies, including electron microscopy, light scattering techniques, fluorescence microscopy that suggest formation of fibrils.

It is not clear in which ways this peptide scaffold differs from other previously reported self-assembling peptides and why this would find other use beyond previously described applications such as e.g. for vaccines, materials, scaffolds with increased catalytic efficiency or guest protein assemblies.

The main novelty is demonstration of selfassembly inside mammalian cells. This has been used to demonstrate the effect of Nck clusters on actin polymerization. This is not a novel finding as the stoichiometry of the assembly had been determined previously but might provide a useful tool, particularly if this process could be regulated externally, which has not been done before.

An interesting aspect of the report is on formation of condensates. The authors conclude that Y15 should not be positioned at the C-terminus although it is unexpected that the addition of an HA tag at the C-terminus evades this problem since the HA tag is very small. Formation of condensates is mentioned with references to liquid phase separation, however FRAP results suggest that they likely form immobile aggregates, therefore the term condensates may not be most appropriate.

I don't think this contribution stringly contributes to our understanding or application of supramolecular assemblies in cells.

Our response:

Thank you for the important comments and advice. We carefully take the comments into consideration on amending our manuscript as shown in below. All the revision we made are highlighted in yellow in the revised manuscript.

Comment 1

The approach is very similar to the reports by Collier group (e.g. Hudalla et al, Q11 peptide, KFE8 etc), in selection of an amphipathic beta-sheet propensity peptide containing high fraction of aromatic residues. Also attachment of the cargo fluorescent protein is similar.

Comment 2

It is not clear in which ways this peptide scaffold differs from other previously reported self-assembling peptides and why this would find other use beyond previously described applications such as e.g. for vaccines, materials, scaffolds with increased catalytic efficiency or guest protein assemblies.

Our response:

In this study, we designed and used Y15 peptide, which is driven from the Y9 peptide design. Our group has already demonstrated the Y9-based biomaterials such as hydrogel scaffold for cell culture so far (Fukunaga et al. *Pept. Sci.* **100**, 731-737 (2013); Tsutsumi et al., *Bioorg. Med. Chem.* **26**, 3126-3132 (2018); Fukunaga et al. *Bull. Chem. Soc. Jpn.* **92**, 391-399 (2019)). Our group has shown the incorporation of bioactive molecules into a hydrogel in these papers to improve cell attachment and induce cell differentiation. Therefore, as the reviewer comments, here we used the peptide scaffold previously described applications. Actually, we tested other designed peptides, RADA16 peptide, which has also demonstrated biomaterials (Yokoi et al., *PNAS* **102**, 8414-8419 (2005)). However, RADAn-sfGFP (n = 12, 16, 20) is uniformly distributed in cells and showed the same level of fluorescence anisotropy as sfGFP, suggesting that the sequence is not favorable for intracellular assembly (Figure A). Our team is performing fundamental studies to reveal the relationship between peptide sequence and assembling propensity in cells.

Figure A. Monomeric dispersion of the RADA-tagged sfGFPs. **a** CLSM observation of RADA-tagged sfGFPs in HEK293 cells. Scale bar, 20 μ m. Even though peptide was extended to 20 residues, RADA20 (M(RADA)₅-sfGFP), any fluorescent granules were not observed. **b** Fluorescence anisotropy measurements of transfected living HEK293 cells. Error bars = s.d. ($n = 3$).

Comment 3

The main novelty is demonstration of selfassembly inside mammalian cells. This has been used to demonstrate the effect of Nck clusters on actin polymerization. This is not a novel finding as the stoichiometry of the assembly had been determine previously of but might provide a useful tool, particularly if this process could be regulated externally, which has not been done before.

Our response:

One of the critical advantages of self-assembling peptides is high design flexibility. This paper demonstrates that SAP can be used in the intracellular environment and proposes a new intracellular building block. Therefore, it is expected that new SAP-tag technology will be developed based on this paper, such as a self-assembling system responding to external stimuli such as metal ion, enzymatic reaction, and so on. As this reviewer points out, the Nck cluster's density-dependency has indeed been demonstrated by antibody techniques. However, in the cytosol, where antibodies are inaccessible, our Y15-based platform using extremely minimal tags is effective.

Comment 4

An interesting aspect of the report is on formation of condensates. The authors conclude that Y15 should not be positioned at the C-terminus although it is unexpected that the addition of an HA tag at the C-terminus evades this problem since the HA tag is very small.

Our response:

The sfGFP-Y15, which terminates at the Y15 sequence, had a lower fluorescence anisotropy than sfGFP, but its assembly tendency was weaker than the other positions. Considering that assembly was improved by only fusing a small HA tag to the C-terminus, it is suggested that minor differences in the C-terminus sequence affect self-assembly, but we do not know the precise reason. At the same time, this aspect also means that it is not a major biological application problem. Even if the Y15 tag must be fused to the C-terminus of the target protein, we can construct protein assembly by fusing Y15-HA to the C-terminus.

Comment 5

Formation of condensates is mentioned with references to liquid phase separation, however FRAP results suggest that they likely form immobile aggregates, therefore the term condensates may not be most appropriate.

Our response:

As this reviewer suggested, our Y15-granule is a gel-like structure with low protein mobility. Therefore, we deleted the description and reference about liquid phase separation in the introduction. As reviewer #3 mentioned, “protein condensates” are defined as structures driven by phase separation. We substituted the term ‘condensates’ into ‘assembly,’ ‘aggregate,’ ‘structure’ or ‘scaffold’ appropriately.

Modification in the maintext: (modified sites were highlighted)

In page 6, line 20: We substituted “*microscale condensates*” into “*microscale structures*”

In page 8, line 27: We substituted “*protein condensates*” into “*micrometer-sized protein assemblies*”

In page 8, line 29: We substituted “*100-nm-scaled condensates*” into “*100-nm-scaled structure*”

In page 9, line 4: We substituted “*Y15-AG condensate*” into “*Y15-AG scaffold*”

In page 9, line 25: We substituted “*Nck-incorporated condensates*” into “*Nck-incorporated assemblies*”

In page 12, line 1: We substituted “*the condensates*” into “*the assemblies*”

Comment 6

I don't think this contribution stringly contributes to our understanding or application of supramolecular assemblies in cells.

Our response:

As demonstrated in this paper, our system can rationally construct a protein assembly by fusing small peptides (15 residues) equivalent to epitope tags (2xHA, 19 residues). Also, the Y15 platform can rationally control assembly composition, making it useful for biological applications compared with other methods. Above all, the demonstration that de novo peptides are applicable in cells is central in this paper. We do believe that this manuscript must be a

pioneering report that will lead to developing peptide-based tools for various purposes, such as stimulus-responsive assembling peptides. Thus, we think that the scientific importance of this paper is high.

Response to Reviewer 3's comments

Comments

In the manuscript by Miki and colleagues, the authors investigate the use of short peptides that contain alternating hydrophobic and charged residues creating function cellular compartments. They find that a short, 15 residue peptide (Y15) readily self-assembles into structures both in vitro and in cells. The authors then demonstrate that the structures formed by their peptide can be functionalized. Using an Nck signaling pathways to the actin cytoskeleton, the authors show that these structures are able to promote local actin polymerization.

This study provides a potentially valuable tool for manipulating cellular function using self-assembling platforms linked to specific signaling pathways or other cellular functions. However, this reviewer found that the manuscript was a bit confusing because of the invocation of phase separation and then stating that the structures aren't phase separated. This confusion can be cleared up by performing the proper experiments that can specifically show whether these structures are phase separated or not; these are described below. Importantly, understanding the biophysics of structure formation is essential for promoting this experimental platform, regardless of whether these structures form through phase separation or another mechanism like polymerization. This author also found the description of actin polymerization by Nck and the dissection of the signaling pathway to be lacking; suggestions are also included below. However, because of the novelty of this particular self-assembling platform and its potential usefulness to a broad range of readers, if the authors are able to address this reviewer's concerns, this manuscript will be suitable for publication in Nature Communications.

Major Comments:

1) The authors switch between phase separation and non-phase separated structures throughout the manuscript. In the introduction, they refer to phase separation, yet their results may indicate that the Y15 structures are not phase separated, leaving this review confused as to why phase separation was invoked early on in the paper as the mechanism pointed to for self-assembly. Depending on the results of the experiments in Major Comments 2, draft the manuscript to focus on phase separation or self-assembly into an oligomer, not both.

2) The authors do not definitely show the mechanism of self-assembly and should perform experiments to clarify either phase separation or oligomer formation. FRAP of cellular condensates isn't indicative of the mechanism of formation, rather of the dynamics of molecules. Some structures initially undergo liquid-liquid phase separation and then mature into gel-like condensates (Lin et al. Mol Cell 2015) and would have similar dynamics as what is observed in the cellular experiments here. Because valency is important (smaller peptides do not self-assemble and increasing the valency of AG from monomer to tetramer increases the propensity to form cellular structures), phase separation is a possible mechanism that underlies the formation of these structures. The authors should perform phase behavior experiments in which they 1) increase the concentration of Y15 peptide in their buffer of choice to determine if there is a critical concentration above which structures form and 2)

increase salt concentration to determine if the structure assembly is regulated by salt concentration or increase / decrease the temperature at which their assays are performed to determine if there is a temperature dependence for

structure assembly. Because their peptide forms structures through a combination of hydrophobic interactions, pi-pi interactions, and electrostatic interactions, changing the salt concentration would be informative; at low salt, pi-pi and electrostatic interactions would drive phase separation and at high salt, hydrophobic interactions would drive phase separation.

3) The experiments in which the authors investigate a requirement for nuclear localization shouldn't involve the membrane. If these structures form through phase separation, targeting them to a membrane, will change the dimensionality of system and the phase behavior. These experiments should be repeated using a nuclear export sequence to target the peptides to the cytoplasm, not the plasma membrane.

4) Nck-induced actin polymerization needs to be clarified. Nck-dependent actin polymerization has been studied extensively in cells (Rivera et al, Curr Biol 2004, Ditlev et al J Cell Biol 2012, Taslimi et al Nat Comms 2014). It is the phase separation of Nephrin, Nck, N-WASP that controls the stoichiometry resulting in optimal actin polymerization (Case et al Science 2019 provides a lens into the mechanism that regulates the results in Ditlev et al 2012 and Su et al Science 2016), not just oligomerization. Because phase separation is the mechanism by which Nck promotes actin polymerization through N-WASP and the Arp2/3 complex, the experiments in which the first SH3 domain and Linker are deleted (both of which contribute to phase separation (Banjade et al PNAS 2015) and the double Nck support phase separation as the mechanism driving local actin polymerization. In these scenarios, the authors alter the valency of Nck and thus likely the propensity of phase separate. The authors also state that by varying the amount of Nck in condensates, actin polymerization changes. This isn't entirely accurate, as Ditlev et al J Cell Biol 2012 clearly showed that the amount of Nck in clusters isn't the driver of actin polymerization, density and stoichiometry with N-WASP and Arp2/3 complex are the driving factors. No additional experiments are needed. Rather, the authors experiments should be considered in light of these previous studies and this section should be rewritten.

Minor Comments:

1) Line 4, 'Microtubes' should be microtubules.

2) Line 5, 'Biomolecular condensates, or protein phase separation' aren't analogous terms. Phase separation can drive biomolecular condensate formation. This should be fixed.

3) Figure 1D should include representative EM images of each peptide at the condition shown for Y15, if the images are available.

4) It appears that the structure of the Y15 and Y15-sfGFP are different. Could the authors comment on why? Does the sfGFP alter the structural layout or might the Y15 interact with the surface of sfGFP to induce different structures?

5) The method used to quantify the Y15 content of the cellular structures may not be the best or easiest way to accomplish this. The authors should repeat analysis of the images and measure partition coefficient (Fluorescence inside structures vs. outside structure) to quantify the differences between Y15, Y13, and other Y peptides as applicable (See Banani et al Cell 2016 for partition coefficient information).

Our response:

We appreciate the reviewers' valuable and important comments and advice. According to the comments, we ran the experiments and answered the reviewer's concerns one by one below. We carefully consider their comments on amending our manuscript. All the revisions we made are highlighted in yellow in the revised manuscript.

Comment 1

1) The authors switch between phase separation and non-phase separated structures throughout the manuscript. In the introduction, they refer to phase separation, yet their results may indicate that the Y15 structures are not phase separated, leaving this review confused as to why phase separation was invoked early on in the paper as the mechanism pointed to for self-assembly. Depending on the results of the experiments in Major Comments 2, draft the manuscript to focus on phase separation or self-assembly into an oligomer, not both.

Our response:

We agree with the reviewer's opinion that after identifying whether the assembly of Y15 peptide is a phase separation or an oligomer formation, the introduction should focus on either. Therefore, additional experiments were performed on the self-assembly of Y15. The results are described later in Comment 2. Based on these results, we concluded that the mechanism of Y15 assembly is not the phase separation but most likely oligomer formation. Therefore, in the introduction section, the reference and description of phase separation were deleted.

Modification in the maintext: (modified sites were highlighted)

In page 3, line 5: The sentence “*Biomolecular condensates, or protein phase separation, integrate subsets of proteins to accelerate or attenuate reactions and signaling processes²⁻⁵*” in the original manuscript was deleted. The revised manuscript is shown below.

“In natural biological systems, non-covalent interactions drive macromolecules to spontaneously organize into supramolecular structures, giving rise to complex functionality. For example, cytoskeleton components, such as actin filaments and microtubules, which play central roles in cell shaping, migration, and intracellular transportation, form in this manner¹. Therefore, platform techniques that build artificial protein assemblies in living systems are crucial for achieving the ultimate goal of synthetic biology: elucidating biological systems through mimicry and generating novel systems with artificial functions^{2,3}. “

Comment 2

2) The authors do not definitely show the mechanism of self-assembly and should perform experiments to clarify either phase separation or oligomer formation. FRAP of cellular condensates isn't indicative of the mechanism of formation, rather of the dynamics of molecules. Some structures initially undergo liquid-liquid phase separation and then mature into gel-like condensates (Lin et al. Mol Cell 2015) and would have similar dynamics as what is observed in the cellular experiments here. Because valency is important (smaller peptides do not self-assemble and increasing the valency of AG from monomer to tetramer increases the propensity to form cellular structures), phase separation is a possible mechanism that underlies the formation of these structures. The authors should perform phase behavior experiments in which they 1) increase the concentration of Y15 peptide in their buffer of choice to determine if there

is a critical concentration above which structures form and 2)

increase salt concentration to determine if the structure assembly is regulated by salt concentration or increase / decrease the temperature at which their assays are performed to determine if there is a temperature dependence for structure assembly. Because their peptide forms structures through a combination of hydrophobic interactions, pi-pi interactions, and electrostatic interactions, changing the salt concentration would be informative; at low salt, pi-pi and electrostatic interactions would drive phase separation and at high salt, hydrophobic interactions would drive phase separation.

Our response:

The reviewer requires to clarify whether the mechanism of Y15 self-assembly is phase separation or oligomer formation. According to the reviewer's comment, for this purpose, we studied two experiments: (1) the peptide concentration dependence of the phase behavior and (2) the salt strength or temperature dependence for structure assembly to reveal the driving interaction. For each, answer as follows.

(1) Peptide concentration dependency of the Y15 phase behavior

We have already examined the concentration-dependent fibril formation of Y15 peptide by measuring the thioflavin-T (ThT) fluorescence intensity and confirmed that the Y15 peptide self-assembles at the detection limit of 5 μ M. However, this method is for evaluating the fibrils formation, not for observing the phase behavior. Furthermore, after overnight incubation at 37 °C, measurements were carried out. As Reviewer 3 pointed out, in order to investigate the possibility of fibril formation through LLPS, the concentration-dependent assembly process of Y15 peptide was assessed by imaging with differential interference contrast (DIC) microscopy and measuring the time-course of ThT fluorescence intensity. Surprisingly, ThT fluorescence enhanced without a lag time and reached to plateau within 3 min (Figure B-a). In contrast, any structure was not observed initially, then ThT-positive fibrillar aggregates arose and elongated (Figure B-b and c). Although the time required to form the aggregates was concentration-dependent, aggregates were observed even at 3 μ M after 72 hours of incubation. Since liquid droplets, spherical structures, were not detected at any incubation time, we expect that Y15 peptides do not undergo phase separation but rapidly self-assemble into fibrils that gradually entangle into aggregates. TEM observation revealed that the microscale aggregates consisted of entangled fibers (Fig. S3, newly added).

Figure B. Properties of the Y15 peptide assembling. **a**, Time-course of thioflavin-T fluorescence intensity. To 25 μM of thioflavin-T solution in PBS, DMSO stock of Y15 peptide (4 mM) was added (final concentration is 20 μM). The fluorescence intensity was measured by F-7000 (Hitachi). Excitation, 430 nm (10 nm slit); Emission, 490 nm (20 nm slit). **b**, DIC images of Y15 aggregation. Scale bar, 50 μm . **c**, Thioflavin-T fluorescence image of Y15 aggregate. Thioflavin-T positive aggregates were clearly observed at all condition (3, 5, 10 and 20 μM of Y15). Scale bar, 50 μm .

Events in the low concentration range (μM or less) cannot be observed by DIC or by ThT assays due to the detection limit. We have already studied the low concentration range of Y15-sfGFP. From our fluorescence anisotropy measurements (Fig. S9), Y15-sfGFP shifts from the monomer to oligomer state at 50 -200 nM. However, no aggregate was observed in this concentration range. On the other hand, Y15-sfGFP was accumulated in fibril structures at 5 μM solution (Fig. 2e), and granular fluorescence was observed under a microscope in PBS. Based on these results, it is considered that at 50 nM and above, the Y15 peptide forms oligomer, and at μM and above, the fiber-like assemblies collide and entangle to form insoluble aggregates that can be observed with an optical microscope. This trend is similar to the self-assembly of Amyloid β 42. A β 42 is known to form oligomers (Sci. Rep. 8, 1783 (2018)) when the

concentration exceeds 90 nM of CAC (critical aggregation concentration). Meanwhile, soluble A β 42 decreased, and the aggregate started to form at concentrations exceeding 0.2 μ M (ACS neurosci. 1, 13 -18 (2010)).

Taken together, the mechanism of Y15 self-assembly is thought to be the formation of oligomers without involving liquid-liquid phase separation, and the interaction of oligomers (fibrils) forms aggregates at higher concentrations. Therefore, we deleted the description on LLPS and focused on the protein assembly (oligomer and fibril) in the introduction of our manuscript. The above results were added to the supporting information (Fig. S4, newly added).

(2) Salt concentration or temperature dependency of the Y15 assembly

The reviewer requires experiments of the salt concentration or temperature dependence of Y15 self-assembly to assess the driving interaction. We have already examined the difference in salt strength dependence between PBS and 10 mM sodium phosphate buffer with low salt strength and described it in the original paper. However, to accurately respond to the reviewer's comments and clarify these effects, we observed Y15 phase behaviors by DIC microscopy, and ThT fluorescence intensities were also measured. By varying the NaCl concentration (0, 150, 300, 500, 1,000 mM) in 10 mM sodium phosphate buffer, we assessed the ionic strength dependency of Y15 self-assembly. The fluorescence intensity of ThT saturated within a few minutes in all conditions and was not significantly different at 0-500 mM NaCl but decreased at a higher salt concentration (1 M NaCl) (Fig. C-a). In the DIC observation, aggregates were observed at any conditions, and these aggregates were all ThT-positive, and the remarkable difference could not be confirmed (Fig. C-b). We speculate that electrostatic interaction less contributes to peptide self-association than hydrogen bonding of the main chain and hydrophobic interaction of tyrosine residues, due to the following two reasons: (1) The affinity increased in a chain length-dependent manner despite different net charges among Y9-15 peptides, and (2) the self-assembling was less susceptible to ion strength below 500 mM NaCl condition. However, since ThT fluorescence decreased as increasing salt intensity, electrostatic interaction may have contributed to forming the cross- β structure to some extent. The above results were added to the supporting information (Fig. S5, newly added), and the description of the paper was changed.

Prior to amyloid formation, α -Synuclein undergo the UCST type phase separation, in which the proteins disassemble by heating (Ray et al, Nat. Chem. 12, 705-716 (2020)). To check the effect of temperature on Y15 assembly, we heated the Y15 solution and observed it by DIC microscopy. After heating at 80 $^{\circ}$ C for 90 min, the aggregates became larger and precipitated at the bottom. Thus, high temperature promoted Y15 aggregation instead of dissolving the aggregates. In Y15, the hydrophobic interaction is strengthened due to the dehydration by the heating, and it seems to facilitate aggregate formation. The results were added to the supporting information (Fig. S4d, newly added).

Figure C. Effect of ionic strength or temperature on Y15 assembling. **a** Thioflavin-T fluorescence intensity of Y15 assembly in different ionic strength. To 25 μ M of thioflavin-T solution in 10 mM sodium phosphate buffer (pH 7.4) with different concentration of NaCl (0, 150, 300, 500 and 1,000 mM), DMSO stock of Y15 peptide (2 mM) was added (final concentration is 10 μ M). The fluorescence intensity was measured by F-7000 (Hitachi). Excitation, 430 nm (10 nm slit); Emission, 490 nm (20 nm slit). **b** Thioflavin-T fluorescence image of Y15 aggregates. Thioflavin-T positive aggregates were clearly observed at all condition (0, 150, 300, 500 and 1,000 mM NaCl). **c** High temperature promoting Y15 aggregation. The 3D pictures show that Y15 precipitated at the bottom after heating (80 °C for 90 min), whereas Y15 aggregates are floating before heating.

Modification in the maintext: (modified sites were highlighted)

In page 5, line 26 (Correspond to comment (1 and 2)): We added the following description.

“Some intrinsically disordered regions undergo liquid-liquid phase separation (LLPS) and subsequently form fibril structures²¹⁻²³. To investigate whether Y15 self-assembles through phase separation, we examined the time profile of thioflavin-T fluorescence intensity and observed the phase behavior by differential interference contrast (DIC) microscopy (Fig. S4). Surprisingly, enhancement of thioflavin-T fluorescence occurred without a lag time and plateaued within 3 min. In contrast, no structure was observed initially; then thioflavin-T-positive fibrillar aggregates arose and elongated. The time required for aggregate formation was concentration dependent. The aggregate grew and precipitated when heated to 80 °C. Because liquid droplets were not detected at any incubation time, we expected

Y15 to self-assemble rapidly into fibrils that gradually entangle to form aggregates. TEM observations revealed that the microscale aggregates consisted of entangled fibers (Fig. S3). “

In page 6, line 4 (Correspond to comment (2)): We revised the description of the ion strength effects on Y15 assembly.

“Although the net charges of the Yn peptides at physiological pH differed (Y9 and Y13, neutral; Y11 and Y15, -1), their ability to self-assemble was easily improved by increasing the length. Y15 assembly was less susceptible to ionic strength below 500 mM NaCl (Fig. S5). We hypothesize that electrostatic interactions between Glu and Lys side chains contribute less to assembly than hydrogen bonds on the main chain and the hydrophobic/aromatic interactions of Tyr on the hydrophobic face.”

Figure S3c: We added different magnitudes of EM images of Y15 assemblies.

Figure S4: We added the results of the properties of the Y15 assembling including (a) time-course of ThT intensity, (b) DIC images of Y15 aggregation, (c) ThT fluorescence image of Y15 aggregate, and (d) High temperature promoting Y15 aggregation.

Figure S5: We added the effect of ionic strength on Y15 assembling, including (a) ThT intensity and (b) ThT fluorescence images of Y15 aggregate in different ionic strength.

Comment 3

3) The experiments in which the authors investigate a requirement for nuclear localization shouldn't involve the membrane. If these structures form through phase separation, targeting them to a membrane, will change the dimensionality of system and the phase behavior. These experiments should be repeated using a nuclear export sequence to target the peptides to the cytoplasm, not the plasma membrane.

Our response:

As the reviewer suggested, we performed experiments. Y15-sfGFP-NES in which nuclear export sequence was fused at the C-terminus was expressed in cells. As a result, in highly expressing cells, Y15-sfGFP-NES was observed as single large fluorescent granules exceeding 5 μm of diameter in the cytosol (Fig. D-a). This directly shows that nuclear localization is not necessary for Y15 self-assembling. Besides of this experiment, we performed FRAP analysis to check the dynamics because we noticed that the structure is relatively spherical, a typical feature of LLPS. These granules showed no fluorescence recovery indicating rigid solid-like structures (Fig. D-b).

As described later paragraph, additional protein-protein interactions are required for constructing micrometer scale structures. Y15-sfGFP-NES granules were thought to be formed by a cooperative interaction of both Y15 fibril formation and the sfGFP dimerization. In particular, the Y15-fused tetrameric AG exhibited structures in which smaller aggregates were gathered, whereas the Y15-dimeric sfGFP formed a large gel with uniform fluorescence intensity in the structure. It is suggested that the valency of protein-protein interactions changes the granular shape and size. Meanwhile, Y15-monomeric protein, mAG or mCherry did not form granular structures but exist as oligomer states. The protein-protein interaction is supposed to cross-link fibrils to form large networks.

Figure D. Granular formation of Y15-sfGFP-NES in cytosolic space. **a** CLSM images of Y15-sfGFP-NES expressing HEK293 cells. Transfected HEK293 cells were stained with Hoechst 33342 and observed by CLSM. Scale bar, 10 μm . **b**, FRAP analysis of the Y15-sfGFP-NES granule in HEK293 cells. The fluorescence intensity in the bleached region was not recovered within 2 min. Scale bar, 2 μm .

Modification in the maintext: (modified sites were highlighted)

In page 7, line 25: We added description about Y15-sfGFP-NES assembly.

“We observed that most of the Y15-sfGFP **granules** were localized in the nucleus but not in the nucleolus (Fig. S11). To investigate whether nuclear localization is necessary for assembly, **we tested the assembly of Y15-sfGFP-NES (nuclear export sequence) in the cytosolic space. Y15-sfGFP-NES integrated into single micrometer-sized fluorescent granules in each cell (Fig. 3f), whereas cells expressing Y15-sfGFP-NES at low levels exhibited small puncta (Fig. S11). Fluorescence recovery after photobleaching (FRAP) experiments revealed that the granules are gel-like aggregates without fluidity, which is distinct from that of LLPS (Fig. S11).**”

Figure 3f: We added the results of Y15-sfGFP-NES assemblies in HEK293 cells.

Figure S11b and c: We added the results of (b) Y15-sfGFP-NES assemblies in HEK293 cells and (c) FRAP analysis of Y15-sfGFP-NES.

Comment 4

4) Nck-induced actin polymerization needs to be clarified. Nck-dependent actin polymerization has been studied extensively in cells (Rivera et al, Curr Biol 2004, Ditlev et al J Cell Biol 2012, Taslimi et al Nat Comms 2014). It is the phase separation of Nephrin, Nck, N-WASP that controls the stoichiometry resulting in optimal actin polymerization (Case et al Science 2019 provides a lens into the mechanism that regulates the results in Ditlev et al 2012 and Su et al Science 2016), not just oligomerization. Because phase separation is the mechanism by which Nck promotes actin polymerization through N-WASP and the Arp2/3 complex, the experiments in which the first SH3 domain and Linker are deleted (both of which contribute to phase separation (Banjade et al PNAS 2015) and the double Nck support phase separation as the mechanism driving local actin polymerization. In these scenarios, the

authors alter the valency of Nck and thus likely the propensity of phase separate. The authors also state that by varying the amount of Nck in condensates, actin polymerization changes. This isn't entirely accurate, as Ditlev et al J Cell Biol 2012 clearly showed that the amount of Nck in clusters isn't the driver of actin polymerization, density and stoichiometry with N-WASP and Arp2/3 complex are the driving factors. No additional experiments are needed. Rather, the authors experiments should be considered in light of these previous studies and this section should be rewritten.

Our response:

We appreciate the valuable comments. As the reviewer suggested, we rewrote the description. The major changes were the following five points.

(1) We referred previous reports demonstrating Nck-clustering in living cells (Rivera et al, Curr Biol 2004, Ditlev et al J Cell Biol 2012, Taslimi et al Nat Comms 2014).

(2) We added the description about stoichiometry models which have been proposed by Mayer and colleagues (Ditlev et al J Cell Biol 2012) and about phase separation underlying the mechanism (Banjade et al PNAS 2015, Case et al Science 2019).

(3) We added the description of the previous report on multivalency of SH3 domain promoting phase separation, and we mentioned that our results consistent with these reports.

(4) In the original manuscript, we used the term “dose”. However, the term is not accurate and should be “density”. Mayer and colleagues demonstrated that high Nck-density is critical for actin polymerization by varying proportion of Nck protein in co-aggregates on membranes (Ditlev et al, JCB 2012). Similar to the antibody-based methods, in our report, the effect of the density of Nck on actin polymerization in the cytosolic space were investigated by Y15-based co-assembling. Therefore, we changed the term from dose-dependency to density-dependency for clarity.

(5) The results obtained in this study were not consistent with the antibody-based methods. Although our results show a bell-shaped response to Nck density in actin polymerization, Ditlev et al shows the nonlinear enhancement of actin polymerization as increasing Nck density. On the other hand, our results were consistent with those of in vitro experiments, in which purified N-WASP, Nck, Arp 2/3, and actin were mixed. Taken together, we speculate that the molecular mechanisms may differ between events on the plasma membrane and those in cytosol or solution. For example, it has been reported that PIP2 is involved in the activity of N-WASP, but a complex with such a lipid molecule cannot be expected in vitro or in cytosol. In addition, it is feasible that the difference of dimensionality (from 2D to 3D) affects N-WASP activation system. We clearly described the differences from the antibody-based methods and explanation about the presumed reasons.

Modification in the maintext: (modified sites were highlighted)

In page 9, line 17 (Correspond to comment (1) and (2)): We referred the previous reports about Nck clustering in cells and about phase separation mechanism.

“The clustering of Nck SH3 domains and subsequent actin polymerization has also been demonstrated in living cells by using antibody-based systems³⁷ and CRY2oligo optogenetic clustering tools³⁸. The Nck/N-WASP/Arp2/3 complex model was proposed by comparison of computational simulations with experimental results³⁹. Moreover, recent studies^{35,40} have revealed that phase separation driven by multivalent interactions between Nck and N-WASP is the

*mechanism responsible for polymerization, and the relative stoichiometry of the components regulates the N-WASP dwell time, which correlates with actin polymerization*³⁵. “

In page 9, line 32 (Correspond to comment (3)): We referred the previous reports about multivalency of Nck-WASP interaction.

“In vitro structure-function analysis showed that the linker between the first two SH3 domains of Nck promotes phase separation and allosterically activates N-WASP^{36,40}. *Increasing the SH3 domain valency has been shown to enhance the N-WASP dwell time and concomitant actin assembly*³⁵. *To verify the contribution of the sequence and domain valency to actin polymerization in cellular experiments, we tested a Nck(109–258) truncation mutant containing the latter two SH3 domains (SH3B and SH3C) and a tandemly repeated 2×Nck(1–258) construct (Fig. 5c).”*

In page 4, line 14 (Correspond to comment (4)): We substituted the term “dose” into “density”. We add the term “valency”.

“From in-cell reconstitution studies, the contribution of domain valency and the density-dependency on function were evaluated in situ.”

In page 10, line 9 (Correspond to comment (4)): We substituted the term “dose” into “density”.

“Finally, we tested the Nck density-dependency on actin polymerization. One advantage of self-assembling peptides is compositional control of assembly. Because the assembling moiety is a single component, the monomer’s ratio directly reflects the compositions of co-assemblies. The plasmid concentration correlates linearly with the proportion of Y15-mCherry-Nck(1–258) in Y15-AG co-assemblies (Fig. S21), facilitating the density-dependency assay. The results exhibited a bell-shaped relationship between the Nck proportion and F-actin intensity (Fig. 5d and S22).”

In page 10, line 14 (Correspond to comment (5)): We added explanation about the difference and similarity between our results and previous reports.

*“Similar results have been reported for in vitro experiments using purified full-length N-WASP. In this report, the high concentration of Nck attenuates N-WASP activation, suggesting that excess Nck binding to WASP impairs the promotion of nucleation*⁴². *These results, however, are inconsistent with reported data using antibody-induced aggregation methods, which showed that a high density of Nck is critical for actin polymerization on the membrane*³⁹. *We hypothesize that the molecular mechanism responsible for actin polymerization differs between events on the membrane and those in the cytosol or solution because PIP2 (phosphatidylinositol 4,5-biphosphate) regulates N-WASP activation*⁴³, *and Nck is involved in the activation of N-WASP on PIP2-induced vesicles*⁴⁴.”

Comment 5

1) Line 4, ‘Microtubes’ should be microtubules.

Our response: According to the referee’s comment, we corrected the word.

Modification in the maintext:

In page 3, line 4: We substituted “microtubes” into “microtubules”

Comment 6

2) Line 5, 'Biomolecular condensates, or protein phase separation' aren't analogous terms. Phase separation can drive biomolecular condensate formation. This should be fixed.

Our response:

Because the mechanism of Y15 self-assembly is not phase separation, we deleted the sentence in our introduction.

Modification in the maintext:

In page 3, line 5: We deleted the sentence.

Comment 7

3) Figure 1D should include representative EM images of each peptide at the condition shown for Y15, if the images are available.

Our response:

We attached EM images of Y11 and Y13 in Figure 1d. In case of Y9 peptide, any structures were not observed.

Modification in the maintext:

In page 5, line 22: We described about EM images of Y9, Y11 and Y13 peptides.

"Y11 and Y13 peptides also adopted fibril structures but no structure was observed for the shortest Y9 peptide (Fig. 1d)."

Figure 1d: EM images of Y11 and Y13 were attached.

Comment 8

4) It appears that the structure of the Y15 and Y15-sfGFP are different. Could the authors comment on why? Does the sfGFP alter the structural layout or might the Y15 interact with the surface of sfGFP to induce different structures?

Our response:

The fiber width of Y15-sfGFP obtained from the EM image is consistent with the anti-parallel model (Fig. S8). Therefore, we assume that the assembled structure is similar to that of Y15 peptide. However, presumably because of the steric hindrance of sfGFPs, Y15-sfGFP assembly does not form a linear, infinitely elongated structure as observed in Y15 peptide, but instead form a curly, finite length fiber. This description was added to the text.

Modification in the maintext:

In page 6, line 30: We added explanation about the morphological difference between Y15 peptide assemblies and Y15-sfGFP assemblies.

"In contrast, synthetic Y15 peptides formed straight, long fibrils whose ends could not be determined by TEM analysis. The Y15-sfGFP fibrils were curly and the length (119 ± 71 nm) was finite, presumably because of frustrated growth²⁵ driven by the steric constraints of sfGFPs."

Comment 9

5) The method used to quantify the Y15 content of the cellular structures may not be the best or easiest way to accomplish this. The authors should repeat analysis of the images and measure partition coefficient (Fluorescence inside structures vs. outside structure) to quantify the differences between Y15, Y13, and other Y peptides as applicable (See Banani et al Cell 2016 for partition coefficient information).

Our response:

As the reviewer suggested, we analyzed and measured the partition coefficients by the method written in Banani et al *Cell* 2016. Since this report does not describe about the detail of thresholding, we set the thresholds as follows:

Cell regions (Bulk or granule) were identified by percentile (0.8) thresholding on each image. Regions showing more than twice the average value of the cell regions were identified as granules. Cell region subtracted by granule region were set as bulk region. The ratio values fluorescence inside of structures versus bulk region were calculated for partition coefficient. For cells without granules, we set the partition coefficient value to 1. The following figure (Fig. E-a) is the thresholding results of Y15-sfGFP and sfGFP.

Modification in the maintext:

Figure 3b: We substituted Figure 3b into the following Figure E-b.

In page 7, line 12: “Figure 3b shows the **partition coefficients**, indicating that Y15 is required for protein assembly in cells.”

In page 16, line 23: We described this procedure in the “Methods” section

Figure E Analysis of partition coefficient. a, thresholding of images for identification of cell region and granule region. **b**, Partition coefficient of Yn-sfGFP (n = 100 cells). * p-value (< 0.0001), two-sided t-test.

REVIEWERS' COMMENTS

Reviewer #3 (Remarks to the Author):

The authors of this manuscript have performed a sizable amount of work to improve their manuscript and have adequately addressed the concerns of this reviewer. Their additional experiments, quantitative analysis of their microscopy images, and improved manuscript text improve the clarity of their study and strengthen their conclusions. I especially appreciate the distinction made between molecular assembly of the peptide that they study and phase separation as well as their description of Nck-dependent actin polymerization. Because of these revisions, this reviewer recommends the manuscript from Miki et al. be published in Nature Communications.